# Combining magnetoencephalography with magnetic resonance imaging enhances learning of surrogate-biomarkers

**Denis A Engemann[1,2]\*, Oleh Kozynets[1], David Sabbagh[1,3,4], Guillaume Lemaître[1], Gael Varoquaux[1], Franziskus Liem[5], Alexandre Gramfort[1]**

[1]Université Paris-Saclay, Inria, CEA, Palaiseau, France; [2]Department of Neurology, Max Planck Institute for Human Cognitive and Brain Sciences, Leipzig, Germany; [3]Inserm, UMRS-942, Paris Diderot University, Paris, France; [4]Department of Anaesthesiology and Critical Care, Lariboisière Hospital, Assistance Publique Hôpitaux de Paris, Paris, France; [5]University Research Priority Program Dynamics of Healthy Aging, University of Zürich, Zürich, Switzerland

**Abstract** Electrophysiological methods, that is M/EEG, provide unique views into brain health. Yet, when building predictive models from brain data, it is often unclear how electrophysiology should be combined with other neuroimaging methods. Information can be redundant, useful common representations of multimodal data may not be obvious and multimodal data collection can be medically contraindicated, which reduces applicability. Here, we propose a multimodal model to robustly combine MEG, MRI and fMRI for prediction. We focus on age prediction as a surrogate biomarker in 674 subjects from the Cam-CAN dataset. Strikingly, MEG, fMRI and MRI showed additive effects supporting distinct brain-behavior associations. Moreover, the contribution of MEG was best explained by cortical power spectra between 8 and 30 Hz. Finally, we demonstrate that the model preserves benefits of stacking when some data is missing. The proposed framework, hence, enables multimodal learning for a wide range of biomarkers from diverse types of brain signals.

\*For correspondence:
denis-alexander.engemann@inria.fr

## Introduction

Non-invasive electrophysiology assumes a unique role in clinical neuroscience. Magneto- and electophencephalography (M/EEG) have an unparalleled capacity for capturing brain rhythms without penetrating the skull. EEG is operated in a wide array of peculiar situations, such as surgery (*Baker et al., 1975*), flying an aircraft (*Skov and Simons, 1965*) or sleeping (*Agnew et al., 1966*). Unlike EEG, MEG captures a more selective set of brain sources with greater spectral and spatial definition (*Ahlfors et al., 2010*; *Hari et al., 2000*). Yet, neither of them is optimal for isolating anatomical detail. Clinical practice in neurology and psychiatry, therefore, relies on additional neuroimaging modalities with enhanced spatial resolution such as magnetic resonance imaging (MRI), functional MRI (fMRI), or positron emission tomography (PET). Recently, machine learning has received significant interest in clinical neuroscience for its potential to predict from such heterogeneous multimodal brain data (*Woo et al., 2017*). Unfortunately, the effectiveness of machine learning in psychiatry and neurology is constrained by the lack of large high-quality datasets (*Woo et al., 2017*; *Varoquaux, 2017*; *Bzdok and Yeo, 2017*; *Engemann et al., 2018*) and comparably limited understanding about the data generating mechanisms (*Jonas and Kording, 2017*). This, potentially,

**eLife digest** How old are you? What about your body, and your brain? People are used to answering this question by counting the years since birth. However, biological age could also be measured by looking at the integrity of the DNA in cells or by measuring the levels of proteins in the blood. Whether one goes by chronological age or biological age, each is simply an indicator of general health – but people with the same chronological age may have different biological ages, and vice versa.

There are different imaging techniques that can be used to study the brain. A method called MRI reveals the brain's structure and the different types of tissue present, like white and grey matter. Functional MRIs (fMRIs for short) measure activity across different brain regions, while electrophysiology records electrical signals sent between neurons. Distinct features measured by all three techniques – MRI, fMRI and electrophysiology – have been associated with aging. For example, differences between younger and older people have been observed in the proportion of grey to white matter, the communication between certain brain regions, and the intensity of neural activity.

MRIs, with their anatomical detail, remain the go-to for predicting the biological age of the brain. Patterns of neuronal activity captured by electrophysiology also provide information about how well the brain is working. However, it remains unclear how electrophysiology could be combined with other brain imaging methods, like MRI and fMRI. Can data from these three techniques be combined to better predict brain age?

Engemann et al. designed a computer algorithm stacking electrophysiology data on top of MRI and fMRI imaging to assess the benefit of this three-pronged approach compared to using MRI alone. Brain scans from healthy people between 17 and 90 years old were used to build the computer model. The experiments showed that combining all three methods predicted brain age better. The predictions also correlated with the cognitive fitness of individuals. People whose brains were predicted to be older than their years tended to complain about the quality of their sleep and scored worse on memory and speed-thinking tasks.

Crucially, Engemann et al. tested how the algorithm would hold up if some data were missing. This can happen in clinical practice where some tests are required but not others. Positively, prediction was maintained even with incomplete data, meaning this could be a useful clinical tool for characterizing the brain.

limits the advantage of complex learning strategies proven successful in purely somatic problems (*Esteva et al., 2017*; *Yoo et al., 2019*; *Ran et al., 2019*).

In clinical neuroscience, prediction can therefore be pragmatically approached with classical machine learning algorithms (*Dadi et al., 2019*), expert-based feature engineering and increasing emphasis on surrogate tasks. Such tasks attempt to learn on abundant high-quality data an outcome that is not primarily interesting, to then exploit its correlation with the actual outcome of interest in small datasets. This problem is also known as transfer learning (*Pan and Yang, 2009*) which, in its simplest form, is implemented by reusing predictions from a surrogate-marker model as predictors in the small dataset. Over the past years, predicting the age of a person from its brain data has crystalized as a surrogate-learning paradigm in neurology and psychiatry (*Dosenbach et al., 2010*). First results suggest that the prediction error of models trained to learn age from brain data of healthy populations provides clinically relevant information (*Cole et al., 2018*; *Ronan et al., 2016*; *Cole et al., 2015*) related to neurodegenerative anomalies, physical and cognitive decline (*Kaufmann et al., 2019*). For simplicity, this characteristic prediction error is often referred to as the brain age delta $\Delta$ (*Smith et al., 2019*). Can learning of such a surrogate biomarker be enhanced by combining expert-features from M/EEG, fMRI and MRI?

Research on aging has suggested important neurological group-level differences between young and elderly people: Studies have found alterations in grey matter density and volume, cortical thickness and fMRI-based functional connectivity, potentially indexing brain atrophy (*Kalpouzos et al., 2012*) and decline-related compensatory strategies. Peak frequency and power drop in the alpha band (8–12 Hz), assessed by EEG, has been linked to aging-related slowing of cognitive processes,

such as the putative speed of attention (*Richard Clark et al., 2004*; *Babiloni et al., 2006*). Increased anteriorization of beta band power (15–30 Hz) has been associated with effortful compensatory mechanisms (*Gola et al., 2013*) in response to intensified levels of neural noise, that is, decreased temporal autocorrelation of the EEG signal as revealed by flatter 1/f slopes (*Voytek et al., 2015*). Importantly, age-related variability in fMRI and EEG seems to be independent to a substantial degree (*Kumral et al., 2020*).

The challenge of predicting at the single-subject level from such heterogenous neuroimaging modalities governed by distinct data-generating mechanisms has been recently addressed with model-stacking techniques (*Rahim et al., 2015*; *Karrer et al., 2019*; *Liem et al., 2017*). *Rahim et al., 2015* enhanced classification in Alzheimer's disease by combining fMRI and PET using a stacking approach (*Wolpert, 1992*), such that the stacked models used input data from different modalities. *Liem et al., 2017* have then applied this approach to age-prediction and found that combining anatomical MRI with fMRI significantly helped reduce errors while facilitating detection of cognitive impairment. This suggests that stacked prediction might also enable combining MRI with electrophysiology. Yet, this idea faces one important obstacle related to the clinical reality of data collection. It is often not practical to do multimodal assessments for all patients. Scanners may be overbooked, patients may not be in the condition to undergo MRI and acute demand in intensive care units may dominate priorities. Incomplete and missing data is, therefore, inevitable and has to be handled to unleash the full potential of multimodal predictive models. To tackle this challenge, we set out to build a stacking model for predicting age from electrophysiology and MRI such that any subject was included if some data was available for at least one modality. We, therefore, call it opportunistic stacking model.

At this point, there are very few multimodal databases providing access to electrophysiology alongside MRI and fMRI. The Leipzig Mind-Brain-Body (LEMON) dataset (*Babayan et al., 2019*) includes high-quality research-EEG with MRI and fMRI for 154 young subjects and 75 elderly subjects. The dataset used in the present study is curated by the Cam-CAN (*Shafto et al., 2014*; *Taylor et al., 2017*) and was specifically designed for studying the neural correlates of aging continuously across the life-span. The Cam-CAN dataset is currently the largest public resource on multimodal imaging with high-resolution electrophysiology in the form of MEG alongside MRI data and rich neuropsychological data for more than 650 healthy subjects between 17 and 90 years. The choice of MEG over EEG may lead to a certain degree of friction with the aging-related literature in electrophysiology, the bulk of which is based on EEG-studies. Fortunately, MEG and EEG share the same classes of neural generators, rendering the aging-related EEG-literature highly relevant for MEG-based modeling. On the other hand, the distinct biophysics of MEG and EEG makes both modalities complementary methods. While EEG captures sources of any orientation, MEG preferentially captures tangential but not radial sources. Compared to EEG, MEG benefits from the magnetic transparency of the skull, which facilitates source localization by reducing the risk of errors due to an incorrect head conductivity model, but also by limiting the large-scale mixing of neural sources. This significantly increases the signal-to-noise ratio for MEG in higher frequencies, rendering it a formidable technique for studying cortical oscillatory activity (*Lehtelä et al., 1997*; *Gobbelé et al., 1998*). MEG is, therefore, an interesting modality in its own right for developing neuro-cognitive biomarkers while its close link with EEG may potentially open the door to translatable electrophysiology markers suitable for massive deployment with clinical EEG.

Our study focuses on the following questions: 1) Can MRI-based prediction of age be enhanced with MEG-based electrophysiology? 2) Do fMRI and MEG carry non-redundant clinically relevant information? 3) What are the most informative electrophysiological markers of aging? 4) Can potential advantages of multimodal learning be maintained in the presence of missing values?

## Results

### Opportunistic prediction-stacking approach

We begin by summarizing the proposed method. To build a model for predicting age from electrophysiology, fMRI and anatomical MRI, we employed prediction-stacking (*Wolpert, 1992*). As in *Liem et al., 2017*, the stacked models, here, referred to different input data instead of alternative models on the same data. We used ridge regression (*Hoerl and Kennard, 1970*) to linearly predict

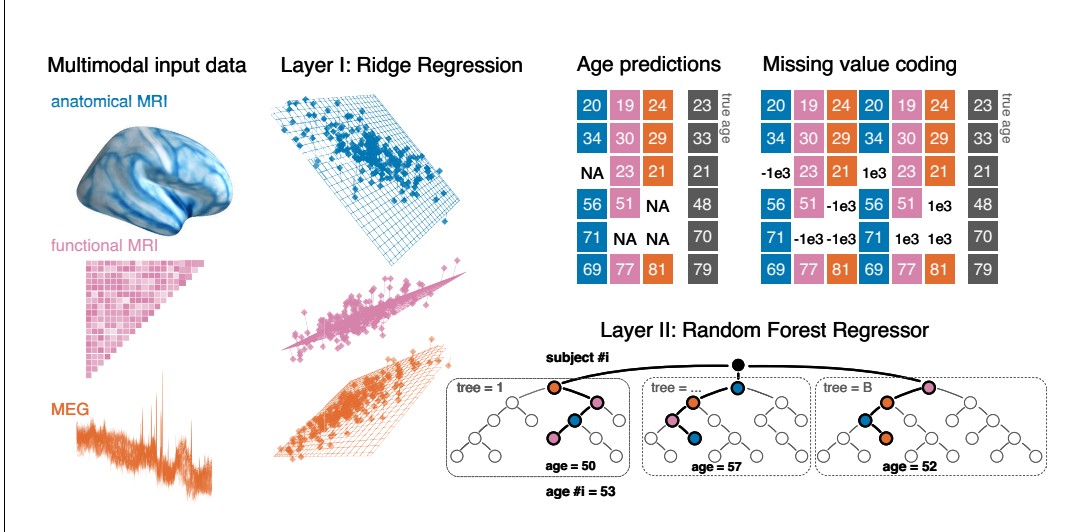

**Figure 1.** Opportunistic stacking approach. The proposed method allows to learn from any case for which at least one modality is available. The stacking model first generates, separately for each modality, linear predictions of age for held-out data. 10-fold cross-validation with 10 repeats is used. This step, based on ridge regression, helps reduce the dimensionality of the data by generating predictions based on linear combinations of the major directions of variance within each modality. The predicted age is then used as derived set of features in the following steps. First, missing values are handled by a coding-scheme that duplicates the second-level data and substitutes missing values with arbitrary small and large numbers. A random forest model is then trained to predict the actual age with the missing-value coded age-predictions from each ridge model as input features. This potentially helps improve prediction performance by combining additive information and introducing non-linear regression on a lower-dimensional representation.

age from high-dimensional inputs of each modality. Linear predictions were based on distinct features from anatomical MRI, fMRI and MEG that have been commonly associated with aging. For extracting features from MEG, in a first step, we drew inspiration from EEG-literature on aging and considered evoked response latencies, alpha band peak frequency, 1/f slope topographies assessed in sensor-space. Previous work on neural development and aging (*Khan et al., 2018*; *Gola et al., 2013*) and Alzheimer's disease (*Gaubert et al., 2019*) has pointed at the importance of spatial alterations in stationary power spectra which can be exploited using high-dimensional regression techniques (*Fruehwirt et al., 2017*). In this work, we have adapted this reasoning to the more general problem of predicting age while exploiting the advanced source-modeling options supported by the Cam-CAN dataset based on MEG and the individual MRIs. Therefore, it was our principal effort to expose the geometry of stationary power spectra with minimal distortion by using source localization based on the individual head geometry (*Sabbagh et al., 2019*) to then perform high-dimensional regression. As a result, we predicted from the spatial distribution of power and bivariate interactions between signals (connectivity) in nine frequency bands (*Table 1*).

For MRI and fMRI, we followed the method established in *Liem et al., 2017* and included cortical thickness, cortical surface area and subcortical volume as well as functional connectivity based on the fMRI time-series. For detailed description of the features, see *Table 2* and section *Feature extraction* in Materials and methods. To correct for the necessarily biased linear model, we then used a non-linear random forest regressor with age predictions from the linear model as lower-dimensional input features.

Thereby, we made sure to use consistent cross-validation splits for all layers and automatically selected central tuning-parameters of the linear model and the random forest with nested cross-validation. Our stacked models handle missing values by treating missing value as data, provided there

**Table 1.** Frequency band definitions.

| Name | Low | $\delta$ | $\theta$ | $\alpha$ | $\beta_1$ | $\beta_2$ | $\gamma_1$ | $\gamma_2$ | $\gamma_3$ |
|---|---|---|---|---|---|---|---|---|---|
| range (Hz) | 0.1 - 1.5 | 1.5 - 4 | 4 - 8 | 8 - 15 | 15 - 26 | 26 - 35 | 35 - 50 | 50 - 74 | 76 - 100 |

**Table 2.** Summary of extracted features.

| # | Modality | Family | Input | Feature | Variants | Spatial selection |
|---|----------|--------|-------|---------|----------|-------------------|
| 1 | MEG | sensor mixed | ERF | latency | aud, vis, audvis | max channel |
| 2 | ... | ... | PSD$\alpha$ | peak | | max channel |
| 3 | ... | ... | PSD | 1/f slope | low, $\gamma$ | max channel in ROI |
| 4 | ... | source activity | signal | power | low,$\delta$,$\theta$,$\alpha$,$\beta_{1,2}$, $\gamma_{1,2,3}$ | MNE, 448 ROIs |
| 5 | ... | ... | envelope | ... | ... | ... |
| 6 | ... | source connectivity | signal | covariance | ... | ... |
| 7 | ... | ... | envelope | ... | ... | ... |
| 8 | ... | ... | env. | corr. | ... | ... |
| 9 | ... | ... | env. | corr. ortho. | ... | ... |
| 10 | fMRI | connectivity | time-series | correlation | ... | 256 ROIs |
| 11 | MRI | anatomy | volume | cortical thickness | | 5124 vertices |
| 12 | ... | ... | surface | cortical surface area | | 5124 vertices |
| 13 | ... | ... | volume | subcortical volumes | | 66 ROIs |

Note. ERF = event related field, PSD = power spectral density, MNE = Minimum Norm-Estimates, ROI = region of interest, corr. = correlation, ortho. = orthogonalized.

is an opportunity to see at least one modality (*Josse et al., 2019*). We, therefore, call it opportunistic stacking model. Concretely, the procedure duplicated all variables and inserted once a small value and once a very large value where data was initially missing for which we chose biologically implausible age values of $-1000$ and $1000$, respectively. For an illustration of the proposed model architecture, see *Figure 1* section *Stacked-Prediction Model for Opportunistic Learning* in Materials and methods for a detailed description of the model.

## fMRI and MEG non-redundantly enhance anatomy-based prediction

Currently, anatomical MRI is the canonical modality for brain age prediction. However, MRI does not access brain dynamics, whereas MEG and fMRI both capture neuronal activity, hence, convey additional information at smaller time-scales. How would they add to the prediction of brain age when combined with anatomical MRI? *Figure 2A* depicts a model comparison in which anatomical MRI served as baseline and which tracked changes in performance as fMRI and MEG were both added through stacking (black boxplot). Anatomical MRI scored an expected generalization error of about 6 years ($SD = 0.6$, $P_{2.5,97.5} = [4.9, 7.16]$), whereas expected chance-level prediction was about 15.5 years ($SD = 1.17$, $P_{2.5,97.5} = [13.26, 17.8]$) based on a dummy-model proposing as prediction the average age of the training-data. MRI performed better than chance-level prediction in every single cross-validation fold. The average improvement over chance-level prediction across folds was at least 9 years ($SD = 1.33$, $P_{2.5,97.5} = [-12.073, -7.347]$). Relative to MRI, age-prediction performance was reduced by almost 1 year on average by adding either MEG ($Pr_{<MRI} = 91\%$, $M = -0.79$, $SD = 0.57$, $P_{2.5,97.5} = [-1.794, 0.306]$) or fMRI ($Pr_{<MRI} = 94\%$, $M = -0.96$, $SD = 0.59$, $P_{2.5,97.5} = [-1.99, 0.15]$). Finally, the performance gain was greater than 1 year on average ($Pr_{<MRI} = 99\%$, $M = -1.32$, $SD = 0.672$, $P_{2.5,97.5} = [-2.43, -0.16]$) when adding both MEG and fMRI to the model, yielding an expected generalization error of about 4.7 years ($SD = 0.55$, $P_{2.5,97.5} = [3.77, 5.74]$). Note that dependable numerical p-values are hard to obtain for paired model comparisons based on cross-validation on the same dataset: Many datasets equivalent to the Cam-CAN would be required. Nevertheless, the uncertainty intervals extracted from the cross-validation distribution suggests that the observed differences in performance were systematic and can be expected to generalize as more data is analyzed. Moreover, the out-of-sample ranking between the different models was stable over cross-validation folds (*Figure 2—figure supplement 1*) with the full model achieving the first rank 71/100 times and performing at least 80/100 better than the MRI + fMRI or the MRI + MEG model. This emphasizes that the relative importance of MEG and fMRI for enhancing MRI-based prediction of age can be expected to generalize to future data.

The improved prediction obtained by combining MEG and fMRI suggests that both modalities carry independent information. If MEG and MRI carried purely redundant information, the random forest algorithm would not have reached better out-of-sample performance. Indeed, comparing the cross-validated prediction errors of MEG-based and fMRI-based models (*Figure 2B*), errors were only weakly correlated ($r_{Spearman} = 0.139$, $r^2 = 0.019$, $p = 1.31 \times 10^{-3}$). fMRI, sometimes, made extreme errors for cases better predicted by MEG in younger people, whereas MEG made errors in distinct cases from young and old age groups. When adding anatomical MRI to each model, the errors became somewhat more dependent leading to moderate correlation ($r_{Spearman} = 0.45$, $r^2 = 0.20$, $p = 2.2 \times 10^{-16}$). This additive component also became apparent when considering predictive simulations on how the model actually combined MEG, fMRI and MRI (*Figure 2—figure supplement 2*) using two-dimensional partial dependence analysis (*Karrer et al., 2019*; *Hastie et al., 2005*, chapter 10.13.2). Moreover, exploration of the age-dependent improvements through stacking suggest that stacking predominantly reduced prediction errors uniformly (*Figure 2—figure supplement 3*) instead of systematically mitigating brain age bias (*Le et al., 2018*; *Smith et al., 2019*).

These findings demonstrate that stacking allows to enhance brain-age prediction by extracting information from MEG, fMRI and MRI while mitigating modality-specific errors. This raises the question whether this additive information from multiple neuroimaging modalities also implies non-redundant associations with behavior and cognition.

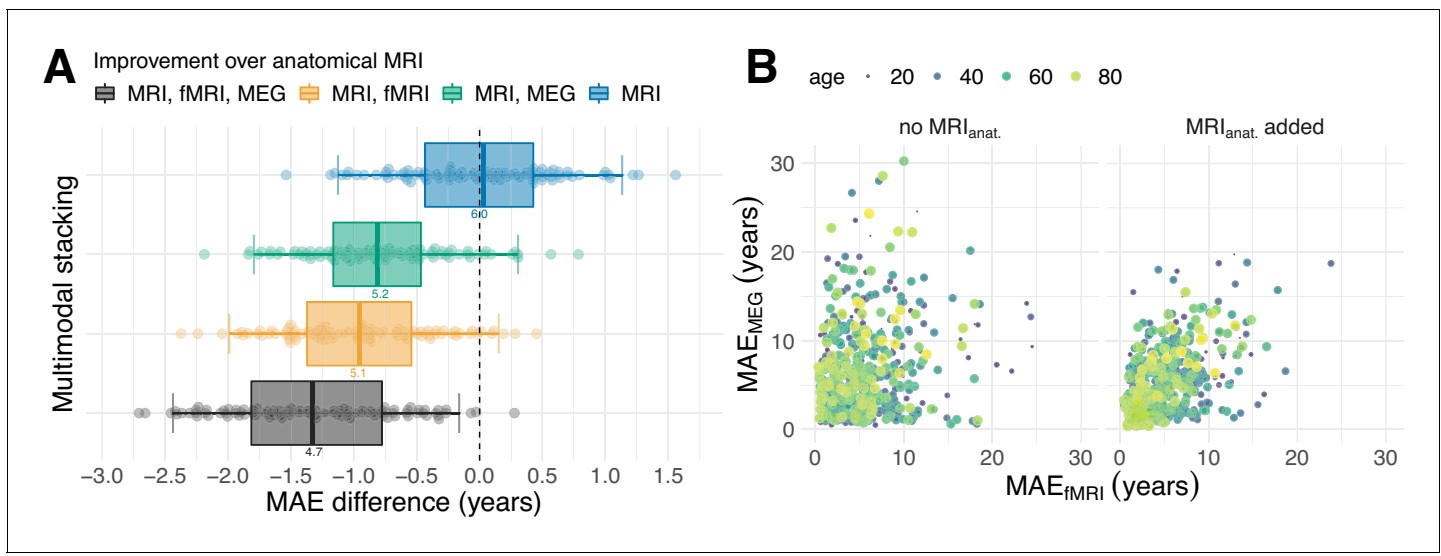

**Figure 2.** Combining MEG and fMRI with MRI enhances age-prediction. (**A**) We performed age-prediction based on distinct input-modalities using anatomical MRI as baseline. Boxes and dots depict the distribution of fold-wise paired differences between stacking with anatomical MRI (blue), functional modalities, that is fMRI (yellow) and MEG (green) and complete stacking (black). Each dot shows the difference from the MRI testing-score at a given fold (10 folds × 10 repetitions). Boxplot whiskers indicate the area including 95% of the differences. fMRI and MEG show similar improvements over purely anatomical MRI around 0.8 years of error. Combining all modalities reduced the error by more than one year on average. (**B**) Relationship between prediction errors from fMRI and MEG. Left: unimodal models. Right: models including anatomical MRI. Here, each dot stands for one subject and depicts the error of the cross-validated prediction (10 folds) averaged across the 10 repetitions. The actual age of the subject is represented by the color and size of the dots. MEG and fMRI errors were only weakly associated. When anatomy was excluded, extreme errors occurred in different age groups. The findings suggest that fMRI and MEG conveyed non-redundant information. For additional details, please consider our supplementary findings.

The online version of this article includes the following figure supplement(s) for figure 2:

**Figure supplement 1.** Rank statistics.
**Figure supplement 2.** Partial dependence.
**Figure supplement 3.** Relationship between predication performance and age.

## Brain age Δ learnt from MEG and fMRI indexes distinct cognitive functions

The brain ageΔ has been interpreted as indicator of health where positive Δ has been linked to reduced fitness or health-outcomes (*Cole et al., 2015*; *Cole et al., 2018*). Does improved performance through stacking strengthen effect-sizes? Can MEG and fMRI help detect complementary associations? *Figure 3* summarizes linear correlations between the brain ageΔ and the 38 neuropsychological scores after projecting out the effect of age, *Equations 6- 8* (see *Analysis of brain-behavior correlation* in Materials and methods for a detailed overview). As effect sizes can be expected to be small in the curated and healthy population of the Cam-CAN dataset, we considered classical hypothesis testing for characterizing associations. Traditional significance testing (*Figure 3A*) suggests that the best stacking models supported discoveries for between 20% (7) and 25% (9) of the scores. Dominating associations concerned fluid intelligence, depression, sleep quality (PSQI), systolic and diastolic blood pressure (cardiac features 1,2), cognitive impairment (MMSE) and different types of memory performance (VSTM, PicturePriming, FamousFaces, EmotionalMemory). The model coefficients in *Figure 3B* depict the strength and direction of association. One can see that stacking models not only tended to suggest more discoveries as their performance improved but also led to stronger effect sizes. However, the trend is not strict as fMRI seemed to support unique discoveries that disappeared when including the other modalities. Similarly, some effect sizes were even slightly stronger in sub-models, for example for fluid intelligence in MRI and MEG. A priori, the full model enjoys priority over the sub-models as its expected generalization estimated with cross-validation was lower. This could imply that some of the discoveries suggested by fMRI may suffer from overfitting, but are finally corrected by the full model. Nevertheless, many of the remaining associations were found by multiple methods (e.g. fluid intelligence, sleep quality assessed by PSQI) whereas others were uniquely contributed by fMRI (e.g. depression). It is also noteworthy that the directions of the effects were consistent with the predominant interpretation of the brain age Δ as indicator of mental or physical fitness (note that high PSQI score indicate sleeping difficulties whereas lower

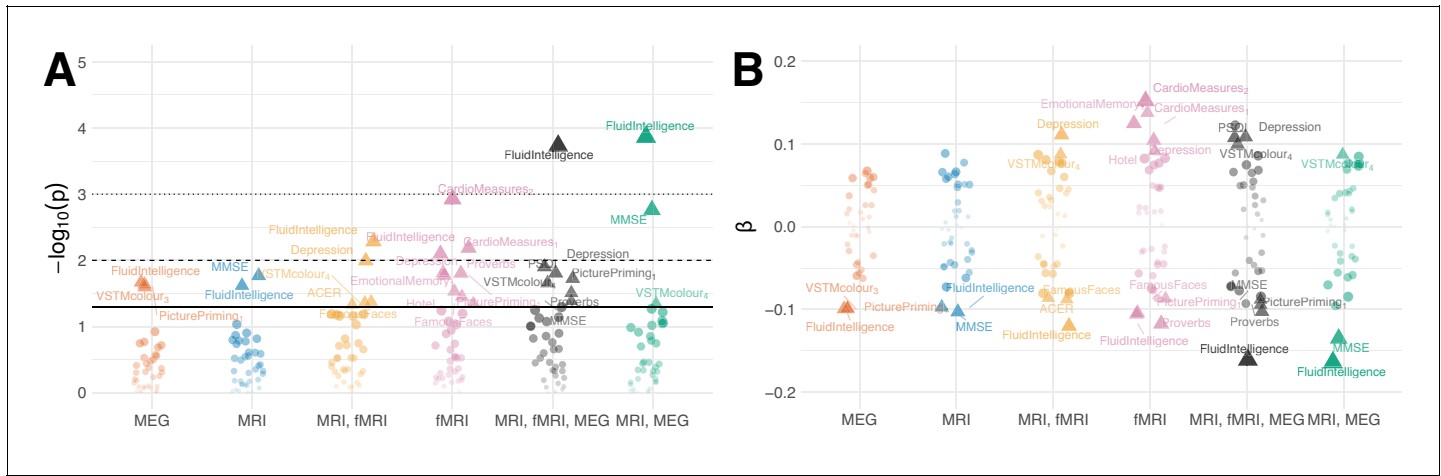

**Figure 3.** Residual correlation between brain ageΔ and neuropsycholgical assessment. (**A**) Manhattan plot for linear fits of 38 neuropsychology scores against brain ageΔ from different models (see scores for *Table 5*). Y-axis: $-log_{10}(p)$. X-axis: individual scores, grouped and colored by stacking model. Arbitrary jitter is added along the x-axis to avoid overplotting. For convenience, we labeled the top scores, arbitrarily thresholded by the uncorrected 5% significance level, indicated by pyramids. For orientation, traditional 5%, 1% and 0.1% significance levels are indicated by solid, dashed and dotted lines, respectively. (**B**) Corresponding standardized coefficients of each linear model (y-axis). Identical labeling as in (**A**). One can see that, stacking often improved effect sizes for many neuropsychological scores and that different input modalities show complementary associations. For additional details, please consider our supplementary findings.

The online version of this article includes the following figure supplement(s) for figure 3:

**Figure supplement 1.** Results based on joint deconfounding.

**Figure supplement 2.** Results based on joint deconfounding with additional regressors of non-interest.

**Figure supplement 3.** Distribution of neuropsychological scores by age.

**Figure supplement 4.** Distribution of neuropsychological scores by age after residualizing.

**Figure supplement 5.** Bootstrap estimates.

MMSE scores indicate cognitive decline) and directly confirm previous findings (*Liem et al., 2017*; *Smith et al., 2019*).

Note that the results were highly similar when performing deconfounding jointly via multiple regression (*Equation 9*, *Figure 3—figure supplement 1*) instead of predicting age-residualized neuropsychological scores, and when including additional predictors of non-interest, that is gender, handedness and head motion (*Equation 10*, *Figure 3—figure supplement 2*). More elaborate confounds-modeling even seemed to improve SNR as suggested by an increasing number of discoveries and growing effect sizes.

These findings suggest that brain age Δ learnt from fMRI or MEG carries non-redundant information on clinically relevant markers of cognitive health and that combining both fMRI and MEG with anatomy can help detect health-related issues in the first place. This raises the question of what aspect of the MEG signal contributes most.

## MEG-based age-prediction is explained by source power

Whether MEG or EEG-based assessment is practical in the clinical context depends on the predictive value of single features, the cost for obtaining predictive features and the potential benefit of improving prediction by combining multiple features. Here, we considered purely MEG-based age prediction to address the following questions: Can the stacking method be helpful to analyze the importance of MEG-specific features? Are certain frequency bands of dominating importance? Is information encoded in the regional power distribution or more related to neuronal interactions between brain regions? *Figure 4A* compares alternative MEG-based models stacking different combinations of MEG-features. We compared models against chance-level prediction as estimated with a mean-regressor outputting the average age of the training data as prediction. Again, chance-level was distributed around 15.5 years ($SD = 1.17$, $P_{2.5,97.5} = [13.26, 17.80]$). All models performed markedly better. The model based on diverse sensor space features from task and resting state recordings showed the lowest performance around 12 years MAE ($SD = 1.04$, $P_{2.5,97.5} = [9.80, 13.52]$), yet it was systematically better than chance ($Pr_{<Chance} = 98.00\%$, $M = -4$, $SD = 1.64$, $P_{2.5,97.5} = [-7.11, -0.44]$). All models featuring source-level power spectra or connectivity ('Source Activity, Source Connectivity') performed visibly better, with expected errors between 8 and 6.5 years and no overlap with the distribution of chance-level scores. Models based on source-level power spectra ('Source Activity', $M = 7.40$, $SD = 0.82$, $P_{2.5,97.5} = [6.01, 9.18]$) and connectivity ('Source Connectivity', $M = 7.58$, $SD = 0.90$, $P_{2.5,97.5} = [6.05, 9.31]$) performed similarly with a slight advantage for the 'Source Activity' model. The best results were obtained when combining power and connectivity features ('Full', $M = 6.75$, $SD = 0.83$, $P_{2.5,97.5} = [5.36, 8.20]$). Adding sensor space features did not lead to any visible improvement of 'Full' over 'Combine Source' with virtually indistinguishable error distributions. The observed average model-ranking was highly consistent over cross-validation testing-splits (*Figure 4—figure supplement 1*), suggesting that the relative importance of the different blocks of MEG features was systematic, hence, can be expected to generalize to future data. The observed ranking between MEG models suggests that regional changes in source-level power spectra contained most information while source-level connectivity added another portion of independent information which helped improve prediction by at least 0.5 years on average. A similar picture emerged when inspecting the contribution of the Layer-I linear models to the performance of the full model in terms of variable importance (*Figure 4B*). Sensor space features were least influential, whereas top contributing features were all related to power and connectivity, which, upon permutation, increased the error by up to 1 year. The most informative input to the stacking model were ridge regression models based on either signal power or the Hilbert analytic signal power concatenated across frequency bands ($P_{cat}$, $E_{cat}$). Other noteworthy contributions were related to power envelope covariance (without source leakage correction) as well as source power in the beta (15–30 Hz) and alpha (8–15 Hz) band frequency range. The results suggest that regional changes in power across different frequency bands are best summarized with a single linear model but additional non-linear additive effects may exist in specific frequency bands. The observed importance rankings were highly consistent with importance rankings obtained from alternative methods for extraction of variable importance (*Figure 4—figure supplement 2*), emphasizing the robustness of these rankings.

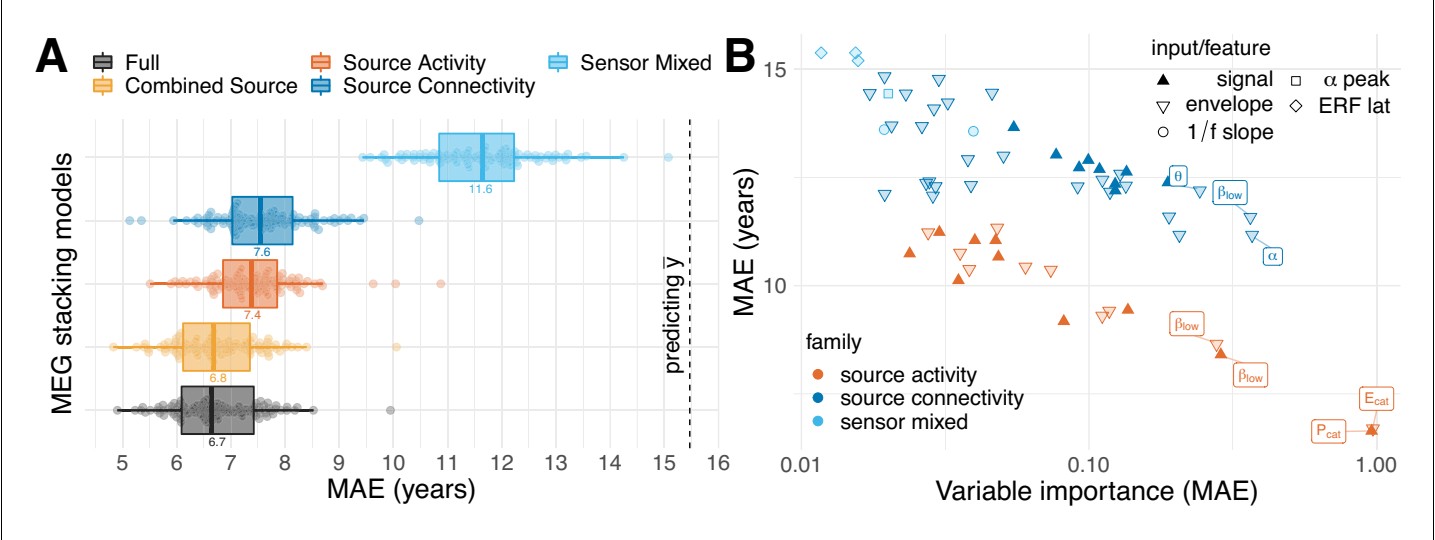

**Figure 4.** MEG performance was predominantly driven by source power. We used the stacking-method to investigate the impact of distinct blocks of features on the performance of the full MEG model. We considered five models based on non-exhaustive combinations of features from three families. 'Sensor Mixed' included layer-1 predictions from auditory and visual evoked latencies, resting-state alpha-band peaks and 1/f slopes in low frequencies and the beta band (sky blue). 'Source Activity' included layer-1 predictions from resting-state power spectra based on signals and envelopes simultaneously or separately for all frequencies (dark orange). 'Source Connectivity' considered layer-1 predictions from resting-state source-level connectivity (signals or envelopes) quantified by covariance and correlation (with or without orthogonalization), separately for each frequency (blue). For an overview on features, see *Table 2*. Best results were obtained for the 'Full' model, yet, with negligible improvements compared to 'Combined Source'. (B) Importance of linear-inputs inside the layer-II random forest. X-axis: permutation importance estimating the average drop in performance when shuffling one feature at a time. Y-axis: corresponding performance of the layer-I linear model. Model-family is indicated by color, characteristic types of inputs or features by shape. Top-performing age-predictors are labeled for convenience (p=power, E = envelope, cat = concatenated across frequencies, greek letters indicate the frequency band). It can be seen that solo-models based on source activity (red) performed consistently better than solo-models based other families of features (blue) but were not necessarily more important. Certain layer-1-inputs from the connectivity family received top-rankings, that is alpha-band and low beta-band covariances of the power envelopes. The most important and best performing layer-1 models concatenated source-power across all nine frequency bands. See *Table 4* for full details on the top-10 layer-1 models. For additional details, please consider our supplementary findings.

The online version of this article includes the following figure supplement(s) for figure 4:

**Figure supplement 1.** Rank statistics.
**Figure supplement 2.** Ranking-stability across methods for variable importance.
**Figure supplement 3.** Partial dependence.
**Figure supplement 4.** Performance of solo- versus stacking-models.

Moreover, partial dependence analysis (*Karrer et al., 2019*; *Hastie et al., 2005*, chapter 10.13.2) suggested that the Layer-II random forest extracted non-linear functions (*Figure 4—figure supplement 3*). Finally, the best stacked models scored lower errors than the best linear models (*Figure 4—figure supplement 4*), suggesting that stacking achieved more than mere variable selection by extracting non-redundant information from the inputs.

These findings show that MEG-based prediction of age is predominantly enabled by power spectra that can be relatively easily accessed in terms of computation and data processing. Moreover, the stacking approach applied to MEG data helped improve beyond linear models by upgrading to non-linear regression.

## Advantages of multimodal stacking can be maintained in populations with incomplete data

One important obstacle for combining signals from multiple modalities in clinical settings is that not all modalities are available for all cases. So far, we have restricted the analysis to 536 cases for which all modalities were present. Can the advantage of multimodal stacking be preserved in the absence of complete data or will missing values mitigate prediction performance? To investigate this question, we trained our stacked model on all 674 cases for which we had the opportunity to extract at

least one feature on any modality, hence, opportunistic stacking (see *Figure 1* and *Table 3* in section *Sample* in Materials and methods). We first compared the opportunistic model with the restricted model on the cases with complete data *Figure 5A*. Across stacking models, performance was virtually identical, even when extending the comparison to the cases available to the sub-model with fewer modalities, for example MRI and fMRI. We then scored the fully opportunistic model trained on all cases and all modalities and compared it to different non-opportunistic sub-models on restricted cases (*Figure 5A*, squares). The fully opportunistic model always out-performed the sub-model. This raises the question of how the remaining cases would be predicted for which fewer modalities were available. *Figure 5B* shows the performance of the opportunistic split by subgroups defined by different combinations of input modalities available. As expected, performance degraded considerably on subgroups for which important features (as delineated by the previous results) were not available. See, for example, the subgroup for which only sensor-space MEG was available. This is unsurprising, as prediction has to be based on data and is necessarily compromised if the features important at train-time are not available at predict-time. One can, thus, say that the opportunistic model operates conservatively: The performance on the subgroups reflects the quality of the features available, hence, enables learning from the entire data.

It is important to emphasize that if missing values depend on age, the opportunistic model inevitably captures this information, hence, bases its predictions on the non-random missing data. This may be desirable or undesirable, depending on the applied context. To diagnose this model-behavior, we propose to run the opportunistic random forest model with the observed missing values as input and observations from the input modalities set to zero. In the current setting, the model trained on missing data indicators performed at chance level ($Pr_{<Chance} = 30.00\%$, $M = 0.65$, $SD = 1.68$, $P_{2.5, 97.5} = [-2.96, 3.60]$), suggesting that the missing values were not informative of age.

## Discussion

We have demonstrated improved learning of surrogate biomarkers by combining electrophysiology as accessed through MEG, functional and anatomical MRI. Here, we have focused on the example of

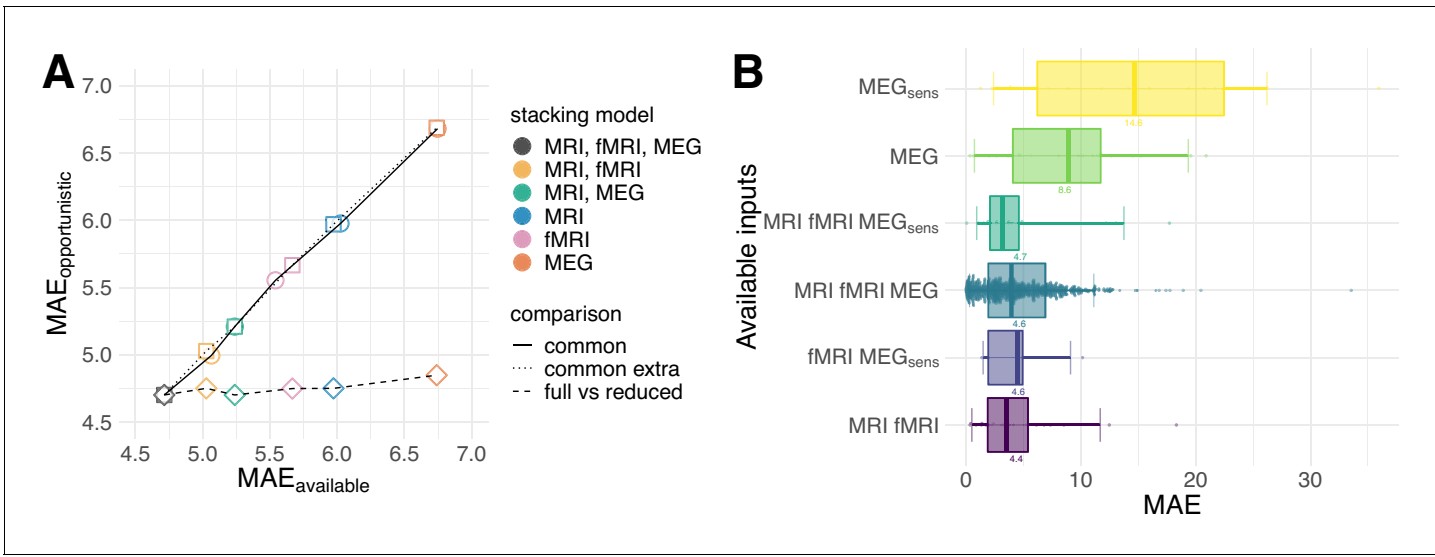

**Figure 5.** Opportunistic learning performance. (**A**) Comparisons between opportunistically trained model and models restricted to common available cases. Opportunistic versus restricted model with different combinations scored on all 536 *common* cases (circles). Same analysis extended to include *extra common* cases available for sub-models (squares). Fully opportunistic stacking model (all cases, all modalities) versus reduced non-opportunistic sub-models (fewer modalities) on the cases available to the given sub-model (diamonds). One can see that multimodal stacking is generally of advantage whenever multiple modalities are available and does not impact performance compared to restricted analysis on modality-complete data. (**B**) Performance for opportunistically trained model for subgroups defined by different combinations of available input modalities, ordered by average error. Points depict single-case prediction errors. Boxplot-whiskers show the 5% and 95% uncertainty intervals. When performance was degraded, important modalities were absent or the number of cases was small, for example, in MEG_sens where only sensor space features were present.

**Table 3.** Available cases by input modality.

| Modality | MEG sensor | MEG source | MRI | fMRI | Common cases |
|---|---|---|---|---|---|
| cases | 589 | 600 | 621 | 626 | 536 |

Note. MEG sensor space cases reflect separate task-related and resting state recordings corresponding to family 'sensor mixed' in **Table 2**. MEG source space cases were exclusively based on the resting state recordings and mapped to family 'source activity' and 'source connectivity' in **Table 2**.

age-prediction by multimodal modeling on 674 subjects from the Cam-CAN dataset, the currently largest publicly available collection of MEG, fMRI and MRI data. Our results suggest that MEG and fMRI both substantially improved age-prediction when combined with anatomical MRI. We have then explored potential implications of the ensuing brain-age Δ as a surrogate-biomarker for cognitive and physical health. Our results suggest that MEG and fMRI convey non-redundant information on cognitive functioning and health, for example fluid intelligence, memory, sleep quality, cognitive decline and depression. Moreover, combining all modalities has led to lower prediction errors. Inspection of the MEG-based models suggested unique information on aging is conveyed by regional distribution of power in the $\alpha$ (8–12 Hz) and $\beta$ (15–30 Hz) frequency bands, in line with the notion of spectral fingerprints (*Keitel and Gross, 2016*). When applied in clinical settings, multimodal approaches should make it more likely to detect relevant brain-behavior associations. We have, therefore, addressed the issue of missing values, which is an important obstacle for multimodal learning approaches in clinical settings. Our stacking model, trained on the entire data with an opportunistic strategy, performed equivalently to the restricted model on common subsets of the data and helped exploiting multimodal information to the extent available. This suggests that the advantages of multimodal prediction can be maintained in practice.

## fMRI and MEG reveal complementary information on cognitive aging

Our results have revealed complementary effects of anatomy and neurophysiology in age-prediction. When adding either MEG or fMRI to the anatomy-based stacking model, the prediction error markedly dropped (*Figure 2A*). Both, MEG and fMRI helped gain almost 1 year of error compared to purely anatomy-based prediction. This finding suggests that both modalities access equivalent information. This is in line with the literature on correspondence of MEG with fMRI in resting state networks, highlighting the importance of spatially correlated slow fluctuations in brain oscillations (*Hipp and Siegel, 2015*; *Hipp et al., 2012*; *Brookes et al., 2011*). On the other hand, recent findings suggest that age-related variability in fMRI and EEG is independent to a substantial degree (*Kumral et al., 2020*; *Nentwich et al., 2020*). Interestingly, the prediction errors of models with MEG and models with fMRI were rather weakly correlated (*Figure 2B*, left panel). In some subpopulations, they even seemed anti-correlated, such that predictions from MEG or fMRI, for the same cases, were either accurate or extremely inaccurate. This additional finding suggests that the improvements of MEG and fMRI over anatomical MRI are due to their access to complementary information that helps predicting distinct cases. Indeed, as we combined MEG and fMRI in one common stacking model alongside anatomy, performance improved on average by 1.3 years over the purely anatomical model, which is almost half a year more precise than the previous MEG-based and fMRI-based models.

These results strongly argue in favor of the presence of an additive component, in line with the common intuition that MEG and fMRI are complementary with regard to spatial and temporal resolution. In this context, our results on performance decomposition in MEG (*Figure 4*) deliver one potentially interesting hint. Source power, especially in the $\alpha(8-15Hz)$ and $\beta(15-26Hz)$ range were the single most contributing type of feature (*Figure 4A*). However, connectivity features, in general, and power-envelope connectivity, in particular, contributed substantively (*Figure 4B*, *Table 4*). Interestingly, applying orthogonalization (*Hipp et al., 2012*; *Hipp and Siegel, 2015*) for removing source leakage did not notably improve performance (*Table 4*). Against the background of research on MEG-fMRI correspondence highlighting the importance of slow fluctuations of brain rhythms (*Hipp and Siegel, 2015*; *Brookes et al., 2011*), this finding suggests that what renders MEG non-redundant with regard to fMRI are regional differences in the balance of fast brain-rhythms, in particular in the $\alpha - \beta$ range.

While this interpretation may be enticing, an important caveat arises from the fact that fMRI signals are due to neurovascular coupling, hence, highly sensitive to events caused by sources other than neuronal activity (*Hosford and Gourine, 2019*). Recent findings based on the dataset analyzed in the present study have shown that the fMRI signal in elderly populations might predominantly reflect vascular effects rather than neuronal activity (*Tsvetanov et al., 2015*). The observed complementarity of the fMRI and MEG in age prediction might, therefore, be conservatively explained by the age-related increase in the ratio of vascular to neuronal contributions to the fMRI signal, while MEG signals are directly induced by neuronal activity, regardless of aging. Nevertheless, in the context of brain-age prediction these mechanisms are less important than the sensitivity of the prediction, for instance, regarding behavioral outcomes.

In sum, our findings suggest that electrophysiology can make a difference in prediction problems in which fast brain rhythms are strongly statistically related to the biomedical outcome of interest.

### Brain age Δ as sensitive index of normative aging

In this study, we have conducted an exploratory analysis on what might be the cognitive and health-related implications of our prediction models. Our findings suggest that the brain age Δ shows substantive associations with about 20–25% of all neuropsychological measures included. The overall big-picture is congruent with the brain age literature (see discussion in *Smith et al., 2019* for an overview) and supports the interpretation of the brain age Δ as index of decline of physical health, well-being and cognitive fitness. In this sample, larger values of the Δ were globally associated with elevated depression scores, higher blood pressure, lower sleep quality, lower fluid intelligence, lower scores in neurological assessment and lower memory performance. Most strikingly, we found that fMRI and MEG contributed additive, if not unique information (*Figure 3*). For example, the association with depression appeared first when predicting age from fMRI. Likewise, the association with fluid intelligence and sleep quality visibly intensified when including MEG.

This extends the previous discussion in suggesting that MEG and fMRI are not only complementary for prediction but also with regard to characterizing brain-behavior mappings. In this context, it is worwhile considering that predicting biomedical outcomes from multiple modalities may reduce susceptibility to 'modality impurity' as often observed in modeling of individual differences in cognitive abilities (*Friedman and Miyake, 2004*; *Miyake et al., 2000*). In the present study, it was remarkable that cardiac measures were exclusively related to fMRI-based models and vanished as MEG was included. This may not be entirely surprising as the fMRI signal is a combination of, both, vascular and neuronal components (*Hosford and Gourine, 2019*) and aging affects both of them differently, which poses an important challenges to fMRI-based studies of aging (*Geerligs et al., 2017*; *Tsvetanov et al., 2016*). It is imaginable that the cardiac measures were not associated with brain age estimates from fMRI when combined with the modalities as vascular components may have enhanced the SNR of neuronal signals through deconfounding (for extensive discussion on this topic, see *Tsvetanov et al., 2019*).

**Table 4.** Top-10 Layer-1 models from MEG ranked by variable importance.

| ID | Family | Input | Feature | Variant | Importance | MAE |
|---|---|---|---|---|---|---|
| 5 | source activity | envelope | power | $E_{cat}$ | 0.97 | 7.65 |
| 4 | source activity | signal | power | $P_{cat}$ | 0.96 | 7.62 |
| 7 | source connectivity | envelope | covariance | $\alpha$ | 0.37 | 10.99 |
| 7 | source connectivity | envelope | covariance | $\beta_{low}$ | 0.36 | 11.37 |
| 4 | source activity | signal | power | $\beta_{low}$ | 0.29 | 8.79 |
| 5 | source activity | envelope | power | $\beta_{low}$ | 0.28 | 8.96 |
| 7 | source connectivity | envelope | covariance | $\theta$ | 0.24 | 11.95 |
| 8 | source connectivity | envelope | correlation | $\alpha$ | 0.21 | 10.99 |
| 8 | source connectivity | envelope | correlation | $\beta_{low}$ | 0.19 | 11.38 |
| 6 | source connectivity | signal | covariance | $\beta_{hi}$ | 0.19 | 12.13 |

Note. ID = mapping to rows from features. MAE = prediction performance of solo-models as in *Figure 4*.

Which neuronal components might explain the enhanced brain-behavior links extracted from the multimodal models? It is enticing to speculate that the regional power of fast-paced $\alpha$ and $\beta$ band brain rhythms captures fast-paced components of cognitive processes such as attentional sampling or adaptive attention (*Gola et al., 2013*; *Richard Clark et al., 2004*), which, in turn might explain unique variance in certain cognitive facets, such as fluid intelligence (*Ouyang et al., 2020*) or visual short-term memory (*Tallon-Baudry et al., 2001*). On the other hand, functional connectivity between cortical areas and subcortical structures, in particular the hippocampus, may be key for depression and is well captured with fMRI (*Stockmeier et al., 2004*; *Sheline et al., 2009*; *Rocca et al., 2015*). Unfortunately, modeling such mediation effects exceeds the scope of the current work, although it would be worth being tested in an independent study with a dedicated design.

Could one argue that the overall effect sizes were too low to be considered practically interesting? Indeed, the strength of linear association was below 0.5 in units of standard deviations of the normalized predictors and the target. On the other hand, it is important to consider that the Cam-CAN sample consists of healthy individuals only. It, thus, appears as rather striking that systematic and neuropsychologically plausible effects can be detected. Our findings, therefore, argue in favor of the brain age Δ being a sensitive marker of normative aging. The effects are expected to be far more pronounced when applying the method in clinical settings, that is, in patients suffering from mild cognitive impairment, depression, neurodevelopmental or neurodegenerative disorders. This suggests that brain age Δ might be used as a screening tool for a wide array of clinical settings for which the Cam-CAN dataset could serve as a normative sample.

## Translation to the clinical setting

One critical factor for application of our approach in the clinic is the problem of incomplete availability of medical imaging and physiological measurements. Here, we addressed this issue by applying an opportunistic learning approach which enables learning from the data available at hand. Our analysis of opportunistic learning applied to age prediction revealed viable practical alternatives to confining the analysis to cases for which all measurements are available. In fact, adding extra cases with incomplete measurements never harmed prediction of the cases with complete data and the full multimodal stacking always outperformed sub-models with fewer modalities (*Figure 5A*). Moreover, the approach allowed maintaining and extending the performance to new cases with incomplete modalities (*Figure 5B*). Importantly, performance on such subsets was explained by the performance of a reduced model with the remaining modalities. Put differently, opportunistic stacking performed as good as a model restricted to data with all modalities. In practice, the approach allows one to improve predictions case-wise by including electrophysiology next to MRI or MRI next to electrophysiology, whenever there is the opportunity to do so.

A second critical factor for translating our findings into the clinic is that, most of the time, it is not high-density MEG that is available but low-density EEG. In this context, our finding showed that the source power was the most important feature, which is of clear practical interest. This is because it suggests that a rather simple statistical object accounts for the bulk of the performance of MEG. Source power can be approximated by the sensor-level topography of power spectra which can be computed on any multichannel EEG device in a few steps and only yields as many variables per frequency band as there are channels. Moreover, from a statistical standpoint, computing the power spectrum amounts to estimating the marginal expectation of the signal variance, which can be thought of as main effect. On the other hand, connectivity is often operationalized as bivariate interaction, which gives rise to a more complex statistical object of higher dimensionality whose precise, reproducible estimation may require far more samples. Moreover, as is the case for power envelope connectivity estimation, additional processing steps each of which may add researcher degrees of freedom (*Simmons et al., 2011*), such as the choice between Hilbert (*Brookes et al., 2011*) versus Wavelet filtering (*Hipp et al., 2012*), types of orthogonalization (*Baker et al., 2014*), and potentially thresholding for topological analysis (*Khan et al., 2018*). This nourishes the hope that our findings will generalize and similar performance can be unlocked on simpler EEG devices with fewer channels. While clinical EEG may not well resolve functional connectivity it may be good enough to resolve changes in the source geometry of the power spectrum (*Sabbagh et al., 2020*). On the other hand, source localization may be critical in this context as linear field spread actually results in a non-linear transform when considering the power of a source (*Sabbagh et al., 2019*). However, in practice, it may be hard to conduct high-fidelity source localization on the basis of low-density EEG and

frequently absent information on the individual anatomy. It will, therefore, be critical to benchmark and improve learning from power topographies in clinical settings.

Finally, it is worthwhile to highlight that, here, we have focused on age in the more specific context of the brain age Δ as surrogate biomarker. However, the proposed approach is fully compatible with any target of interest and may be easily applied directly to clinical end points, for example drug dosage, survival or diagnosis. Moreover, the approach presented here can be easily adapted to work with classification problems, for instance, by substituting logistic regression for ridge regression and by using a random forest classifier in the stacking layer. We have provided all materials from our study in form of publicly available version-controlled code with the hope to help other teams of biomedical researchers to adapt our method to their prediction problem.

## Limitations

For the present study, we see four principal limitations: availability of data, interpretability, nonexhaustive feature-engineering and potential lack of generalizability due to the focus on MEG.

The Cam-CAN is a unique resource of multimodal neuroimaging data with sufficient data points to enable machine learning approaches. Yet, from the point of view of machine learning, the Cam-CAN dataset is a small dataset. This has at least two consequences. If the Cam-CAN included many more data points, for example beyond 10–100 k subjects, the proposed stacking model might possibly be of limited advantage compared to purely non-linear models, for example random forests, gradient boosting or deep learning methods (*Bzdok and Yeo, 2017*). At the same time, the fact that the Cam-CAN has been unique so far, hinders generalization testing to equivalent multimodal datasets from other sites based on alternative scanning methodologies, protocols and devices (*Engemann et al., 2018*). This also renders computation of numerical hypothesis tests (including p-values) more difficult in the context of predictive modeling: The majority of data points is needed for model-fitting and metrics derived from left-out cross-validation splits, for example, predictions of brain age, lack statistical independence. This breaks essential assumptions of inferential statistics to an arbitrary and unknown degree. Our inferences were, therefore, predominantly based on estimated effect-sizes, that is the expected generalization error and its uncertainty assessed through cross-validation.

Second, at this point, statistical modeling faces the dilemma of whether inference or prediction is the priority. Procedures optimizing prediction performance in high dimensions are not yet supported by the in-depth understanding required to guarantee formal statistical inferences, whereas models with well-established procedures for statistical inference lack predictive capability (*Bzdok et al., 2018*; *Bzdok and Ioannidis, 2019*). Forcing interpretation out of machine learning models, therefore, often leads to duplicated analysis pipelines and model specifications, which is undesirable in terms of methodological coherence (for example *Hoyos-Idrobo et al., 2019*; *Haufe et al., 2014*; *Biecek, 2018*). In the present work, we refrained from conducting fine-grained inferential analysis beyond the model comparisons presented, in particular inspection of layer-1 weightmaps whose interpretation remains an ongoing research effort. We hope, nevertheless, that the insights from our work will stimulate studies investigating the link between MEG, fMRI and MRI across the life-span using an inference-oriented framework.

Third, the MEG-features used in the present study were non-exhaustive. Based on the wider MEG/EEG-literature beyond the neuroscience of aging, many other features could have been included. Instead, feature-engineering was based on our aging-specific literature review constrained by biophysical considerations. In particular, the distinction between sensor-space and source-space features was purely descriptive and not substantive. From an empirical perspective, mirroring all features in sensor-space and source-space could have yielded more specific inferences, for example regarding the role of source-power. On the other hand, biophysical prior knowledge implies that oscillatory peak frequencies and evoked response latencies are not modified by source localization, whereas source localization or data-driven approximations thereof are essential for predicting from M/EEG power spectra (*Sabbagh et al., 2019*). It is also fair to admit that, in the present paper, our passion was preferentially attracted by source modeling of neural power spectra. However, one could imagine that with equal investment of resources, more information could have been extracted from the sensor-level features (see *Gemein et al., 2020* for approaches to tackle the important methodological issue of unbalanced investment of development-time). Related, the current work has strongly benefited from expertise on modeling of MEG power spectra under the assumption of

stationary as captured by global power spectra, covariance or connectivity. Recent findings suggest that non-stationary analyses focusing on transient electrophysiological events may uncover clinically relevant information on cognitive brain dynamics (*Barttfeld et al., 2015*; *Baker et al., 2014*; *Vidaurre et al., 2018*; *Van Schependom et al., 2019*). It is, therefore, important to highlight that our proposed framework is open and readily enables integration of additional low- or high-dimensional inputs related to richer sensor-level features or non-stationary dynamics, beyond MEG as input modality.

Finally, while MEG and EEG share the same types of neural generators, their specific biophysics render these methods complementary for studying neuronal activity. At this point, unfortunately, there is no public dataset equivalent of the Cam-CAN including EEG or, both, EEG and MEG. Such a data resource would have enabled studying the complementarity between MEG with EEG as well as generalization from stacking with MRI and MEG to stacking models with MRI and EEG.

We hope that our method will help other scientists to incorporate the multimodal features related to their domain expertise into their applied regression problems.

## Materials and methods

### Sample
We included MEG (task and rest), fMRI (rest), anatomical MRI and neuropsychological data (cognitive tests, home-interview, questionnaires) from the CAM-Can dataset (*Shafto et al., 2014*). Our sample comprised 674 (340 female) healthy individuals between 18 (female = 18) to 88 (female = 87) years with an average of 54.2 (female = 53.7) and a standard deviation of 18.7 (female = 18.8) years. We included data according to availability and did not apply an explicit criterion for exclusion. When automated processing resulted in errors, we considered the data as missing. This induced additional missing data for some cases. A summary of available cases by input modality is reported in *Table 3*. For technical details regarding the MEG, fMRI, and MRI data acquisition, please consider the Cam-CAN reference publications (*Shafto et al., 2014*; *Taylor et al., 2017*).

### Feature extraction
Feature extraction was guided by the perspective of predictive modeling. For the goal of enhancing prediction performance as opposed to statistical inference (*Bzdok and Ioannidis, 2019*), we emphasized on differences between modalities, hence, chose modality-specific methods and optimizations at the risk of sacrificing direct comparability between features used for MEG, fMRI and MRI. The selection of features was guided by our literature review on the neuroscience of aging presented in the introduction.

For MEG, we analyzed sensor space features related to timing (*Price et al., 2017*), peak frequency (*Richard Clark et al., 2004*) and temporal autocorrelation (*Voytek et al., 2015*). Source space features included the power of source-level signals (*Sabbagh et al., 2019*) and envelopes and their bivariate interactions (*Khan et al., 2018*) in nine frequency bands (see *Table 1*, adapted from the Human Connectome Project, *Larson-Prior et al., 2013*). The inclusion of power envelopes was theoretically important as the slow fluctuations of source power and their bivariate interactions have been repeatedly linked to fMRI resting state networks (*Hipp and Siegel, 2015*; *Brookes et al., 2011*). On the other hand, we specifically focused on the unique capacity of MEG to access spatial information induced by fast-paced brain rhythms emerging from regional sources (*King and Dehaene, 2014*; *Stokes et al., 2015*).

For extracting features from MRI and fMRI, we adapted the approach established by *Liem et al., 2017*. For fMRI, we computed bivariate functional connectivity estimates. For MRI, we focused on cortical thickness, cortical surface area and subcortical volumes. An overview on all features used is presented in *Table 2*. In the remainder of this section, we describe computation details.

#### MEG features
##### Peak evoked latency
Sensory processing may slow down in the course of aging (*Price et al., 2017*). Here, we assessed the evoked response latency during auditory, visual and simultaneous audiovisual stimulation (index 1, *Table 2*). For each of the conditions, we first computed the evoked response. Then, we computed

the root-mean-square across gradiometers and looked up the time of the maximum. In total, this yielded three latency values per subject.

### α-band peak frequency

Research suggests that the alpha-band frequency may be lower in older people. Here, we computed the resting-state power spectrum using a Welch estimator (index 2, *Table 2*). Then, we estimated the peak frequency between 6 and 15 Hz on occipito-parietal magnetometers after removing the 1/f trend using a polynomial regression (degree = 15) by computing the maximum power across sensors and looking up the frequency bin. This yielded one peak value per subject.

### 1/f slope

Long-range auto-correlation in neural time-series gives rise to the characteristic 1/f decay of power on a logarithmic scale. Increases of neural noise during aging are thought to lead to reduced auto-correlation, hence a more shallow slope (*Voytek et al., 2015*). We computed the 1/f slope from the Welch power spectral estimates above on all magnetometers using linear regression (index 3, *Table 2*). The slope is given by the $\hat{\beta}$ of the linear fit with the log-frequencies as predictor and the log-power as target. We obtained one estimate for each of the 102 magnetometers, resulting in a 1/f topography. No further reduction was applied.

### Power and connectivity of source-level signals

The cortical generators of the brain-rhythms dominating the power spectrum change across life-span. To predict from the spatial distribution of MEG power spectra, we relied on source-localization to mitigate distortions due to individual head geometry. We adopted the pipeline optimized for high-dimensional regression presented in *Sabbagh et al., 2019* and modeled power spectra in the time-domain based on covariance estimates after bandpass-filtering. We considered nine frequency bands (see *Table 1*), computed bandpass-filtered minimum norm source-estimates and then summarized the source-time courses ROI-wise by the first principal components with alignment to the surface normals using the 'pca_flip' option provided by MNE-Python (*Gramfort et al., 2013*). To mitigate the curse of dimensionality we used a subdivision of the Desikan-Killiany atlas (*Desikan et al., 2006*) comprising 448 ROIs. This set of ROIs proposed by *Khan et al., 2018* for predictive modeling of neurodevelopmental trajectories was specifically designed to generate approximately equal ROI-size to avoid averaging over inhomogeneous regions with distinct leadfield coverage or to avoid averaging over larger regions that may contain multiple sources cancelling each other. Subsequently, we computed the covariance matrix from the concatenated epochs and used the 448 diagonal entries as power estimates (index 4 *Table 2*). The off-diagonal entries served as connectivity estimates. Covariance matrices live in a non-Euclidean curved space. To avoid model violations at the subsequent linear-modeling stages, we used tangent space projection (*Varoquaux et al., 2010*) to vectorize the lower triangle of the covariance matrix. This projection allows one to treat entries of the covariance or correlation matrix as regular Euclidean objects, hence avoid violations to the linear model used for regression (*Sabbagh et al., 2019*). This yielded $448 \times 448/2 - (448/2) = 100,128$ connectivity values per subject (index 6 *Table 2*).

### Power and connectivity of source-level envelopes

Brain-rhythms are not constant in time but fluctuate in intensity. These slow fluctuations are technically captured by power envelopes and may show characteristic patterns of spatial correlation. To estimate power envelopes, for each frequency band, we computed the analytic signal using the Hilbert transform. For computational efficiency, we calculated the complex-valued analytic signal in sensor space and then source-localized it using the linear minimum norm operator. To preserve linearity, we only extracted the power envelopes by taking the absolute value of the analytic signal after having performed averaging inside the ROIs. Once the envelope time-series was computed, we applied the same procedure as for source power (paragraph above) to estimate the source power of the envelopes (index 5, *Table 2*) and their connectivity. Power and covariance were computed from concatenated epochs, correlation and orthogonalized correlation were computed epoch-wise. Note that, for systematic reasons, we also included power estimates of the envelope time-series applying the same method as we used for the time-series. In the MEG literature, envelope correlation is a well-established research topic (*Hipp et al., 2012*; *Brookes et al., 2011*). Thus,

in addtition to the covariance, we computed the commonly used normalized Pearson correlations and orthogonalized Pearson correlations which are designed to mitigate source leakage (index 7–9, *Table 2*). However, as a result of orthogonalization, the resulting matrix is not any longer positive definite and cannot be projected to the tangent space using Riemannian geometry. Therefore, we used Fisher's Z- transform (*Silver and Dunlap, 1987*) to convert the correlation matrix into a set of standard-normal variables. The transform is defined as the inverse hyperbolic tangent function of the correlation coefficient: $z = \operatorname{arctanh}(r) = \frac{1}{2}\log(\frac{1+r}{1-r})$. This yielded 448 power envelope power estimates and 100,128 connectivity values per estimator.

### fMRI features
## Functional connectivity
Large-scale neuronal interactions between distinct brain networks has been repeatedly shown to change during healthy aging. Over the past years, for fMRI-based predictive modeling using functional atlases from about 50 to 1000 ROIs have emerged as a fundamental element for mitigating heterogeneity and dimensionality reduction, especially in small- to medium-sized datasets such as the Cam-CAN with less than 1000 observations (*Dadi et al., 2019*; *Abraham et al., 2017*). To estimate macroscopic functional connectivity, we deviated from the 197-ROI BASC atlas *Bellec et al., 2010* used in *Liem et al., 2017*. Instead, we used an atlas with 256 sparse and partially overlapping ROIs obtained from Massive Online Dictionary Learning (MODL) (*Mensch et al., 2016*). Initial piloting suggested that both methods gave approximately equivalent results on average with slightly reduced variance for the MODL atlas. Then, we computed bivariate amplitude interactions using Pearson correlations from the ROI-wise average time-series (index 10, *Table 2*). Again, we used tangent space projection (*Varoquaux et al., 2010*) to vectorize the correlation matrices. This yielded 32,640 connectivity values from the lower triangle of each matrix. No further reduction was applied.

### MRI features
The extraction of features from MRI followed the previously established strategy presented in *Liem et al., 2017* which is based on cortical surface reconstruction using the FreeSurfer software. For scientific references to specific procedures, see the section *MRI data processing* and the FreeSurfer website http://freesurfer.net/.

## Cortical thickness
Aging-related brain atrophy has been related to thinning of the cortical tissue (for example *Thambisetty et al., 2010*). We extracted cortical thickness, defined as shortest distance between white and pial surfaces, from the Freesurfer (*Fischl, 2012*) segmentation using a surface tessellation with 5124 vertices in fsaverage4 space obtained from the FreeSurfer command `mris_preproc` using default parameters (index 11, *Table 2*). No further reduction was computed.

## Cortical surface area
Aging is also reflected in shrinkage of the cortical surface itself (for example *Lemaitre et al., 2012*). We extracted vertex-wise cortical surface area estimates, defined as average of the faces adjacent to a vertex along the white surface, from the Freesurfer segmentation using a surface tessellation with 5124 vertices in fsaverage4 space obtained from the FreeSufer command `mris_preproc` using default parameters (index 12, *Table 2*). No further reduction was computed.

## Subcortical volumes
The volume of subcortical structures has been linked to aging (for example *Murphy et al., 1992*). Here, we used the FreeSurfer command `asegstats2table`, using default parameters, to obtain estimates of the subcortical volumes and global volume, yielding 66 values for each subject with no further reductions (index 13, *Table 2*).

## Stacked-prediction model for opportunistic learning
We used the stacking framework (*Wolpert, 1992*) to build our predictive model. However, we made the important specification that input models were regularized linear models trained on input data

from different modalities, whereas stacking of linear predictions was achieved by a non-linear regression model. Our model can be intuitively denoted as follows:

$$y = f([X_1\beta_1 \ldots X_m\beta_m]) \tag{1}$$

Here, each $X_j\beta_j$ is the vector of predictions $\hat{y}_j$ of the target vector $y$ from the jth model fitted using input data $X_j$:

$$\{y = X_1\beta_1 + \epsilon_1, \ldots, y = X_m\beta_m + \epsilon_m\} \tag{2}$$

We used ridge regression as input model and a random forest regressor as a general function approximator $f$ [Ch. 15.4.3](*Hastie et al., 2005*). A visual illustration of the model is presented in *Figure 1*.

## Layer-1: Ridge regression

Results from statistical decision theory suggests that, for linear models, the expected out-of-sample error increases only linearly with the number of variables included in a prediction problem (*Hastie et al., 2005*, chapter 2), not exponentially. In practice, biased (or penalized) linear models with Gaussian priors on the coefficients, that is ridge regression (or logistic regression for classification) with $\ell_2$-penalty (squared $\ell_2$ norm) are hard to outperform in neuroimaging settings (*Dadi et al., 2019*). Ridge regression can be seen as extension of ordinary least squares (OLS) where the solution is biased such that the coefficients estimated from the data are conservatively pushed toward zero:

$$\hat{\beta}_{ridge} = (X^\top X + \lambda I)^{-1} X^\top y, \tag{3}$$

The estimated coefficients approach zero as the penalty term $\lambda$ grows, and the solution approaches the OLS fit as $\lambda$ gets closer to zero. This shrinkage affects directions of variance with small singular values more strongly than the ones with large singular values (see eqs. 3.47-3.50 in *Hastie et al., 2005*, ch. 3.4.1), hence, can be seen as smooth principal component analysis as directions of variance are shrunk but no dimension is ever fully discarded. This is the same as assuming that the coefficient vector comes from a Gaussian distribution centered around zero such that increasing shrinkage reduces the variance $\sigma^2$ of that distribution [chapter 7.3] (*Efron and Hastie, 2016*):

$$\beta \sim N\left(0, \frac{\sigma^2}{\lambda}I\right) \tag{4}$$

In practice, the optimal strength for this Gaussian prior is often unknown. For predictive modeling, $\lambda$ is commonly chosen in a data-driven fashion such that one improves the expected out-of-sample error, for example tuned using cross-validation. We tuned $\lambda$ using generalized cross-validation (*Golub et al., 1979*) and considered 100 candidate values on an evenly spaced logarithmic scale between $10^{-3}$ and $10^5$. This can be regarded equivalent to assuming a flat but discrete hyper-prior (a prior distribution of the hyper-parameters assumed for the model parameters) on the distribution of candidate regularization-strengths. Note that this procedure is computationally efficient and, on our problem, returned entire regularization paths within seconds. While this approach is standard-practice in applied machine learning and particularly useful with massive and high-dimensional data, many other methods exist for data-driven choice of the prior which may be more appropriate in situations on smaller datasets and where parameter inference, not prediction, is the priority.

## Layer-2: Random forest regression

However, the performance of the ridge model in high dimensions comes at the price of increasing bias. The stacking model tries to alleviate this issue by reducing the dimensionality in creating a derived data set of linear predictions, which can then be forwarded to a more flexible local regression model. Here, we chose the random forest algorithm (*Breiman, 2001*) which can be seen as a general function approximator and has been interpreted as an adaptive nearest neighbors algorithm (*Hastie et al., 2005*, chapter 15.4.3). Random forests can learn a wide range of functions and are capable of automatically detecting non-linear interaction effects with little tuning of hyper-

parameters. They are based on two principles: regression trees and bagging (bootstrapping and aggregating). Regression trees are non-parametric methods and recursively subdivide the input data by finding combinations of thresholds that relate value ranges of the input variables to the target. The principle is illustrated at the right bottom of *Figure 1*. For a fully grown tree, each sample falls into one leaf of the tree which is defined by its unique path through combinations of input-variable thresholds through the tree. However, regression trees tend to easily overfit. This is counteracted by randomly generating alternative trees from bootstrap replica of the dataset and randomly selecting subset of variables for each tree. Importantly, thresholds are by default optimized with regard to a so-called impurity criterion, for which we used mean squared error. Predictions are then averaged, which mitigates overfitting and also explains how continuous predictions can be obtained from thresholds.

In practice, it is common to use a generous number of trees as performance plateaus once a certain number is reached, which may lay between hundreds or thousands. Here, we used 1000 trees. Moreover, limiting the overall depth of the trees can increase bias and mitigate overfitting at the expense of model complexity. An intuitive way of conceptualizing this step is to think of the tree-depth in terms of orders interaction effects. A tree with three nodes enables learning three-way interactions. Here, we tuned the model to choose between depth-values of 4, 6, or 8 or the option of not constraining the depth. Finally, the total number of features sampled at each node determines the degree to which the individual trees are independent or correlated. Small number of variables de-correlate the trees but make it harder to find important variables as the number of input variables increases. On the other hand, using more variables at once leads to more exhaustive search of good thresholds, but may increase overfitting. As our stacking models had to deal with different number of input variables, we had to tune this parameter and let the model select between $\sqrt{p}$, $\log(p)$ and all $p$ input variables. We implemented selection of tuning-parameters by grid search as (nested) 5-fold cross-validation with the same scoring as used for evaluation of the model performance, that is mean absolute error. The choice of the mean absolute error is a natural choice for the study of aging as error is directly expressed in the units of interest.

## Stacked cross-validation

We used a 10-fold cross-validation scheme. To mitigate bias due to the actual order of the data, we repeated the procedure 10 times while reshuffling the data at each repeat. We then generated age-predictions from each layer-1 model on the left-out folds, such that we had for each case one age-prediction per repeat. We then stored the indices for each fold to make sure the random forest was trained on left-out predictions for the ridge models. This ensured that the input-layer train-test splits where carried forward to layer-2 and that the stacking model was always evaluated on left-out folds in which the input ages are actual predictions and the targets have not been seen by the model. Here, we customized the stacking procedure to be able to unbox and analyze the input-age predictions and implement opportunistic handling of missing values.

## Variable importance

Random forest models and, in general, regression trees are often inspected by estimating the impact of each variable on the prediction performance. This is commonly achieved by computing the so-called variable importance. The idea is to track and sum across all trees the relative reduction of impurity each time a given variable is used to split, hence, the name mean decrease impurity (MDI). The decrease in impurity can be tracked by regular performance metrics. Here we used mean squared error, which is the default option for random forest regression in scikit-learn (*Pedregosa et al., 2011*). It has been shown that in correlated trees, variable importance can be biased and lead to masking effects, that is, fail to detect important variables (*Louppe et al., 2013*) or suggest noise-variables to be important. One potential remedy is to increase the randomness of the trees, for example by selecting randomly a single variable for splitting and using extremely randomized trees (*Geurts et al., 2006*; *Engemann et al., 2018*), as it can be mathematically guaranteed that in fully randomized trees only actually important variables are assigned importance (*Louppe et al., 2013*). However, such measures may mitigate prediction performance or lead to duplicated model specifications (one model for predicting, one for analyzing variable importance). Here, we used the approach from the original random forest paper (*Breiman, 2001*), which consists

in permuting $k$ times one variable at a time and measuring the drop in performance at the units of performance scoring, that is mean absolute error in years. We computed permutation importance with $k = 1000$ after fitting the random forest to the cross-validated predictions from the layer-1 models.

In-sample permutation importance is computationally convenient but may potentially suffer from an irreducible risk of overfitting, even when taking precautions such as limiting the tree-depth. This risk can be avoided by computing the permutations on left-out data, that is by permuting the variables in the testing-set, which can be computationally expensive. However, permutation importance (whether computed on training- or testing -data) has the known disadvantage that it does not capture conditional dependencies or higher order interactions between variables. For example, a variable may not be so important in itself but its interaction with other variables makes it an important predictor. Such conditional dependencies between variables can be captured with MDI importance.

To diagnose potential overfitting and to assess the impact of conditional dependencies, we additionally reported out-of-sample permutation importance and MDI importance. We computed out-of-sample permutation importance for each of the 100 splits from our cross-validation procedure with a reduced number of permutations ($k = 100$) to avoid excessive computation times. MDI importance was based on the same model fit as the in-sample permutations.

## Opportunistic learning with missing values

An important option for our stacking model concerns handling missing values. Here, we implemented the double-coding approach (*Josse et al., 2019*) which duplicates the features and once assigns the missing value a very small and once a very large number (see also illustration in *Figure 1*). As our stacked input data consisted of age predictions from the ridge models, we used biologically implausible values of $-1000$ and $1000$. This amounts to turning missing values into features and let the stacking model potentially learn from the missing values, as the reason for the missing value may contain information on the target. For example, an elderly patient may not be in the best conditions for an MRI scan, but nevertheless qualifies for electrophysiological assessment.

To implement opportunistic stacking, we considered the full dataset with missing values and then kept track of missing data while training layer-1. This yielded the stacking-data consisting of the age-predictions and missing values. Stacking was then performed after applying the feature-coding of missing values. This procedure made sure that all training and test splits were defined with regard to the full cases and, hence, the stacking model could be applied to all cases after feature-coding of missing values.

## Statistical inference

Rejecting a null-hypothesis regarding differences between two cross-validated models is problematic in the absence of sufficiently large unseen data or independent datasets: cross-validated scores are not statistically independent. Fortunately, cross-validation yields useful empirical estimates of the performance (and its dispersion) that can be expected on unseen data (*Hastie et al., 2005*, Ch. 7.10). Here, we relied on uncertainty estimates of paired differences based on the stacked cross-validation with 10 folds and 10 repeats. To provide a quantitative summary of the distributions of paired split-wise differences in performance, we extracted the mean, the standard deviation, the 2.5 and 97.5 percentiles (inner 95% of the distribution) as well as the number of splits in which a model was better than a given reference ($Pr_{<Ref}$). We estimated chance-level prediction using a dummy regressor that predicts the average of the training-set target using the same cross-validation procedure and identical random seeds to ensure split-wise comparability between non-trivial models. While not readily supporting computation of p-values, dummy estimates are computationally efficient and yield distributions equivalent to those obtained from label-permutation procedures. For statistical analyses linking external measurements with model-derived quantities such as the cross-validated age prediction or the brain age $\Delta$, we used classical parametric hypothesis-testing. It should be clear, however, that hypothesis-testing, here, provides a quantitative orientation that needs to be contextualized by empirical estimates of effect sizes and their uncertainty to support inference.

## Analysis of brain-behavior correlation

To explore the cognitive implications of the brain age Δ, we computed correlations with the neuro-behavioral score from the Cam-CAN dataset. *Table 5* lists the scores we considered. The measures fall into three broad classes: neuropsychology, physiology and questionnaires ('Type' columns in *Table 5*). Extraction of neuropsychological scores sometimes required additional computation, which followed the description in *Shafto et al., 2014*, (see also 'Variables' column in scores). For some neuropsychological tasks, the Cam-CAN dataset provided multiple scores and sometimes the final score of interest as described in *Shafto et al., 2014*, had yet to be computed. At times, this amounted to computing ratios, averages or differences between different scores. In other scores, it was not obvious how to aggregate multiple interrelated sub-scores, hence, we extracted the first principal component explaining between about 50% and 85% of variance, hence offering reasonable summaries. In total, we included 38 variables. All neuropsychology and physiology scores (up to #17 in *Table 5*) were the scores available in the 'cc770-scored' folder from release 001 of the Cam-CAN dataset. We selected the additional questionnaire scores (#18-23 in *Table 5*) on theoretical grounds to provide an assessment of clinically relevant individual differences in cognitive functioning. The brain age Δ was defined as the difference between predicted and actual age of the person

$$\text{BrainAge}_{\Delta} = \text{age}_{pred} - \text{age}, \tag{5}$$

**Table 5.** Summary of neurobehavioral scores.

| # | Name | Type | Variables (38) |
|---|------|------|----------------|
| 1 | Benton faces | neuropsychology | total score (1) |
| 2 | Emotional expression recognition | ... | PC1 of RT (1), EV = 0.66 |
| 3 | Emotional memory | ... | PC1 by memory type (3), EV = 0.48,0.66,0.85 |
| 4 | Emotion regulation | ... | positive and negative reactivity, regulation (3) |
| 5 | Famous faces | ... | mean familiar details ratio (1) |
| 6 | Fluid intelligence | ... | total score (1) |
| 7 | Force matching | ... | Finger- and slider-overcompensation (2) |
| 7 | Hotel task | ... | time(1) |
| 9 | Motor learning | ... | M and SD of trajectory error (2) |
| 10 | Picture priming | ... | baseline RT, baseline ACC (4) |
| ... | ... | ... | M prime RT contrast, M target RT contrast |
| 11 | Proverb comprehension | ... | score (1) |
| 12 | RT choice | ... | M RT (1) |
| 13 | RT simple | ... | M RT (1) |
| 14 | Sentence comprehension | ... | unacceptable error, M RT (2) |
| 15 | Tip-of-the-tounge task | ... | ratio (1) |
| 16 | Visual short term memory | ... | K (M,precision,doubt,MSE) (4) |
| 17 | Cardio markers | physiology | pulse, systolic and diastolic pressure 3) |
| 18 | PSQI | questionnaire | total score (1) |
| 19 | Hours slept | ... | total score (1) |
| 20 | HADS (Depression) | ... | total score (1) |
| 21 | HADS (Anxiety) | ... | total score (1) |
| 22 | ACE-R | ... | total score (1) |
| 23 | MMSE | ... | total score (1) |

Note. M = mean, SD = standard deviation, RT = reaction time, PC = principal component, EV = explained variance ratio (between 0 and 1), ACC = accuracy, PSQI = Pittsburgh Sleep Quality Index HADS = Hospital Anxiety and epression Scale, ACE-R = Addenbrookes Cognitive Examination Revised, MMSE = Mini Mental State Examination. Numbers in parentheses indicate how many variables were extracted.

such that positive values quantify overestimation and negative value underestimation. A common problem in establishing brain-behavior correlations for brain age is spurious correlations due to shared age-related variance in the brain age $\Delta$ and the neurobehavioral score (*Smith et al., 2019*). To mitigate confounding effects of age, we computed the age residuals as

$$\text{score}_{resid} = \text{score} - \text{score}_{age}, \tag{6}$$

where $\text{score}$ is the observed neuropsychological score and $\text{score}_{age}$ is its prediction from the following polynomial regression:

$$\text{score}_{age} = \text{age}\beta_1 + \text{age}^2\beta_2 + \text{age}^3\beta_3 + \epsilon, \tag{7}$$

The estimated linear association between the residualized score and the brain age $\Delta$ was given by $\beta_1$ in

$$\text{score}_{resid} = \text{BrainAge}_\Delta\beta_1 + \epsilon, \tag{8}$$

To obtain comparable coefficients across scores, we standardized both the age and the scores. We also included intercept terms in all models which are omitted here for simplicity.

It has been recently demonstrated, that such a two-step procedure can lead to spurious associations (*Lindquist et al., 2019*). We have, therefore, repeated the analysis with a joint deconfounding model where the polynomial terms for age are entered into the regression model alongside the brain age predictor.

$$\text{score} = \text{BrainAge}_\Delta\beta_1 + \text{age}\beta_2 + \text{age}^2\beta_3 + \text{age}^3\beta_4 + \epsilon. \tag{9}$$

Finally, the results may be due to confounding variable of non-interest. To assess the importance of such confounders, we have extended the model (*Equation 9*) to also include gender, handedness (binarized) and a log Frobenius norm of the variability of motion parameters (three translation, three rotation) over the 241 acquired images.

$$\begin{aligned}\text{score}\ \ &= \text{BrainAge}_\Delta\beta_1 + \text{gender}\beta_2 + \text{hand}_{\text{binary}}\beta_3 + \\ &\quad \log(\text{norm}(\text{motion}))\beta_4 + \text{age}\beta_5 + \text{age}^2\beta_6 + \text{age}^3\beta_7 + \epsilon.\end{aligned} \tag{10}$$

Note that motion correction was already performed during preprocessing of MRI and fMRI. Likewise, MEG source localization took into account individual head geometry as well as potentially confounding environmental noise through whitening with the noise covariance obtained from empty room recordings. Following the work by *Liem et al., 2017*, we included total grey matter and total intracranial volume as important features of interest among the MRI-features.

## MEG data processing

### Data acquisition

MEG recorded at a single site using a 306 VectorView system (Elekta Neuromag, Helsinki). This system is equipped with 102 magnetometers and 204 orthogonal planar gradiometers is placed in a light magnetically shielded room. During acquisition, an online filter was applied between around 0.03 Hz and 1000 Hz. This resulted in a sampling-frequency of 1000 Hz. To support offline artifact correction, vertical and horizontal electrooculogram (VEOG, HEOG) as well as electrocardiogram (ECG) signal was concomitantly recorded. Four Head-Position Indicator (HPI) coils were used to measure the position of the head. All types of recordings, that is resting-state, passive stimulation and the active task lasted about 8 min. For additional details on MEG acquisition, please consider the reference publications of the Cam-CAN dataset (*Taylor et al., 2017*; *Shafto et al., 2014*). The following sections will describe the custom data processing conducted in our study.

### Artifact removal

#### Environmental artifacts

To mitigate contamination of the MEG signal with artifacts produced by environmental magnetic sources, we applied temporal signal-space-separation (tSSS) (*Taulu and Kajola, 2005*). The method uses spherical harmonic decomposition to separate spatial patterns produced by sources inside the

head from patterns produced by external sources. We used the default settings with eight components for the harmonic decomposition of the internal sources, and three for the external sources on a ten seconds sliding window. We used a correlation threshold of 98% to ignore segments in which inner and outer signal components are poorly distinguishable. We performed no movement compensation, since there were no continuous head monitoring data available at the time of our study. The origin of internal and external multipolar moment space was estimated based on the head-digitization. We computed tSSS using the MNE `maxwell_filter` function (*Gramfort et al., 2013*) but relied on the SSS processing logfiles from Cam-CAN for defining bad channels.

## Physiological artifacts

To mitigate signal distortions caused by eye-movements and heart-beats we used signal space projection (SSP) (*Uusitalo and Ilmoniemi, 1997*). This method learns principal components on contaminated data-segments and then projects the signal into the sub-space orthogonal to the artifact. To obtain clean estimates, we excluded bad data segments from the EOG/ECG channels using the 'global' option from autoreject (*Jas et al., 2017*). We then averaged the artefact-evoked signal (see 'average' option in `mne.preprocessing.compute_proj_ecg`) to enhance subspace estimation and only considered one single projection vector to preserve as much signal as possible.

## Rejection of residual artifacts

To avoid contamination with artifacts that were not removed by SSS or SSP, we used the 'global' option from autoreject (*Jas et al., 2017*). This yielded a data-driven selection of the amplitude range above which data segments were excluded from the analysis.

## Temporal filtering

To study band-limited brain dynamics, we applied bandpass-filtering using the frequency band definitions in *Table 1*. We used default filter settings from the MNE software (development version 0.19) with a windowed time-domain design (firwin) and Hamming taper. Filter length and transition bandwidth was set using the 'auto' option and depended on the data.

## Epoching

For the active and passive tasks, we considered time windows between −200 and 700 ms around stimulus-onset and decimated the signal by retaining every eighth time sample.

For resting-state, we considered sliding windows of 5 s duration with no overlap and no baseline correction. To reduce computation time, we retained the first 5 min of the recording and decimated the signal by retaining every fifth time sample. Given the sampling frequency of 1000 Hz, this left unaffected the bulk of the features, only reducing the spectral resolution in the high gamma band to 75–100 Hz (instead of 75–120 Hz in the definition proposed by the Human Connectome Project [*Larson-Prior et al., 2013*]).

## Channel selection

It is important to highlight that after SSS, the magnetometer and gradiometer data are reprojected from a common lower dimensional SSS coordinate system that typically spans between 64 and 80 dimensions. After SSS, magnetometers and gradiometers are reconstructed from the same basis vectors, which makes them linear combinations of another (*Taulu and Kajola, 2005*). As a result, both sensor types contain highly similar information and yield equivalent results in many situations (*Garcés et al., 2017*). Consequently, after applying SSS, the MNE software manipulates a single sensor type for source localization and uses as degrees of freedom the number of underlying SSS dimensions instead of the number of channels. Note, however, that after SSS, magnetometers and gradiometers can still yield systematically different results in sensor-space analyses despite being linear combinations of another. This happens once a non-linear transform is applied on the sensor-space, for example power, which is explained by the fact that SSS is a linear transform and powers in sensors space breaks linearity. On the other hand, once source localization is correctly performed, which takes into account the SSS solution, differences between gradiometers and gradiometers become negligible for, both, linear transforms and non-linear transforms. We, nevertheless, used all 102 magnetometers and 204 gradiometers for source analysis to stick with a familiar configuration. Note that while short-cuts can be achieved by processing only one of the sensor types, they should

be avoided when other methods than SSS are used for preprocessing. However, driven by initial visual exploration, for some aspects of feature engineering in sensor space, that is, extraction of alpha peaks or computation of 1/f power spectra, we used the 102 magnetometers. For extraction of evoked response latencies, we used the 204 gradiometers. Nevertheless, due to the characteristic of SSS to combine sensor types into one common representation, in all analyses, magnetic fields sampled from, both, magnetometers and gradiometers were exploited even if only one type of sensors was formally included.

## Covariance modeling

To control the risk of overfitting in covariance modeling (*Engemann and Gramfort, 2015*), we used a penalized maximum-likelihood estimator implementing James-Stein shrinkage (*James and Stein, 1992*) of the form

$$\hat{\Sigma}_{\text{biased}} = (1-\alpha)\hat{\Sigma} + \alpha\frac{\text{Trace}(\hat{\Sigma})}{p}I, \tag{11}$$

where $\alpha$ is the regularization strength, $\hat{\Sigma}$ is the unbiased maximum-likelihood estimator, $p$ is the number of features and $I$ the identity matrix. This, intuitively, amounts to pushing the covariance toward the identity matrix. Here, we used the Oracle Approximation Shrinkage (OAS) (*Chen et al., 2010*) to compute the shrinkage factor $\alpha$ mathematically.

## Source localization

To estimate cortical generators of the MEG signal, we employed the cortically constraint Minimum-Norm-Estimates (*Hämäläinen and Ilmoniemi, 1994*) based on individual anatomy of the subjects. The resulting projection operator exclusively captures inputs from the anatomy of the subject and additional whitening based on the noise covariance. On the other hand, beamforming methods, consider the segments of MEG data to be source-localized through the data covariance. Methods from the MNE-family are therefore also referred to as non-adaptive spatial filters, whereas beamforming methods are referred to as adaptive spatial filters. The MNE-operator can be expressed as

$$W_{\text{MNE}} = G^{\top}(GG^{\top} + \lambda I_P)^{-1}. \tag{12}$$

Here $G \in \mathbb{R}^{P \times Q}$ with $P$ sensors and $Q$ sources denotes the forward model quantifying the spread from sources to M/EEG observations and $\lambda$ a regularization parameter that controls the $\ell_2$-norm of the activity coefficients. This parameter implicitly controls the spatial complexity of the model with larger regularization strength leading to more spatially smeared solutions. The forward model is obtained by numerically solving Maxwell's equations based on the estimated head geometry, which we obtained from the Freesurfer brain segmentation. Note that from a statistical perspective, the MNE-solution is a Ridge model (see *Equations 3-4*) predicting the magnetic field at a given sensor from a linear combination of corresponding entries in the leadfields. The inferred source activity is given by multiplication of the MNE-operator with sensor-level magnetic fields.

We estimated the source amplitudes on a grid of 8196 candidate dipole locations equally spaced along the cortical mantle. We used spatial whitening to approximate the model assumption of Gaussian noise (*Engemann and Gramfort, 2015*). The whitening operator was based on the empty room noise covariance and applied to the MEG signal and the forward model. We applied no noise normalization and used the default depth weighting (*Lin et al., 2006*) as implemented in the MNE software (*Gramfort et al., 2014*) with weighting factor of 0.8 (*Lin et al., 2006*) and a loose-constraint of 0.2. The squared regularization parameter $\lambda^2$ was expressed with regard to the signal-to-noise ratio and fixed at the default value of $\frac{1}{\text{SNR}^2}$ with $\text{SNR} = 3$ for all subjects. This conservative choice was also motivated by the computational burden for optimizing the regularization parameter. Optimizing this hyper-parameter would have required pre-computing hundreds of MNE solutions to then perform grid search over the derived source-level outputs. As the goal was prediction from the source localized signals, not inference on spatial effects, we have instead relied on the subsequent data-driven shrinkage through the level-1 ridge model (see *Equations 3-4*). It may be worthwhile to systematically investigate the interplay between shrinkage at the MNE-level and the ridge-level for predictive modeling with MEG in future research.

## MRI data processing

### Data acquisition

For additional details on data acquisition, please consider the reference publications of the CAM-Can (*Taylor et al., 2017*; *Shafto et al., 2014*). The following sections will describe the custom data processing conducted in our study.

### Structural MRI

For preprocessing of structural MRI data we used the FreeSurfer (version 6.0) software (http://surfer.nmr.mgh.harvard.edu/)) (*Fischl, 2012*). Reconstruction included the following steps (adapted from the methods citation recommended by the authors of FreeSurfer http://freesurfer.net/fswiki/FreeSurferMethodsCitation): motion correction and average of multiple volumetric T1-weighted images (*Reuter et al., 2010*), removal of non-brain tissue (*Ségonne et al., 2004*), automated Talairach transformation, segmentation of the subcortical white matter and deep gray matter volumetric structures (*Fischl et al., 2002*; *Fischl et al., 2004*) intensity normalization (*Sled et al., 1998*), tessellation of the gray-matter/white matter boundary, automated topology correction (*Fischl et al., 2001*; *Ségonne et al., 2004*), and surface deformation following intensity gradients (*Dale et al., 1999*; *Fischl and Dale, 2000*). Once cortical models were computed, so-called deformable procedures were applied including surface inflation (*Fischl et al., 1999*), registration to a spherical atlas (*Fischl et al., 1999*) and cortical parcellation (*Desikan et al., 2006*).

### fMRI

The available fMRI data were visually inspected. The volumes were excluded from the study provided they had severe imaging artifacts or head movements with amplitude larger than 2 mm. After the rejection of corrupted data, we obtained a subset of 626 subjects for further investigation. The fMRI volumes underwent slice timing correction and motion correction to the mean volume. Following that, co-registration between anatomical and function volumes was done for every subject. Finally, brain tissue segmentation was done for every volume and the output data were morphed to the MNI space.

## Scientific computation and software

### Computing environment

For preprocessing and feature-extraction of MEG, MRI and fMRI we used a high-performance Linux server (72 cores, 376 GB RAM) running Ubuntu Linux 18.04.1 LTS. For subsequent statistical modeling, we used a golden Apple MacBook 12″ (early 2016) running MacOS Mojave (8 GB RAM). General purpose computation was carried out using the Python (3.7.3) language and the scientific Python stack including NumPy, SciPy, Pandas, and Matplotlib. For embarrassingly parallel processing, we used the joblib library.

### MEG processing

For MEG processing, we used the MNE-Python software (https://mne.tools) (*Gramfort et al., 2014*) (version 0.19). All custom analysis code was scripted in Python and is shared in a dedicated repository including a small library and scripts (see section Code Availability).

### MRI and fMRI processing

For anatomical reconstruction we used the shell-scripts provided by FreeSurfer (version 6.0) software (*Fischl et al., 2002*). We used the pypreprocess package, which reimplements parts of the SPM12 software for the analysis of brain images (*The Wellcome Centre for Human Neuroimaging, 2018*), complemented by the Python-Matlab interface from Nipype (*Gorgolewski et al., 2011*). For feature extraction and processing related to predictive modeling with MRI and fMRI, we used the NiLearn package (*Abraham et al., 2014*).

### Statistical modeling

For predictive modeling, we used the scikit-learn package (*Pedregosa et al., 2011*) (version 0.21). We used the R (3.5.3) language and its graphical ecosystem (*R Development Core Team, 2019*;

*Wickham, 2016*; *Slowikowski, 2019*; *Clarke and Sherrill-Mix, 2017*; *Canty and Ripley, 2017*) for statistical visualization of data. For computation of ranking-statistics, we used the pmr R-package (*Lee and Yu, 2013*).

### Code availability

We share all code used for this publication on GitHub: https://github.com/dengemann/meg-mri-sur-rogate-biomarkers-aging-2020. (*Engemann, 2020*; https://github.com/elifesciences-publications/meg-mri-surrogate-biomarkers-aging-2020) Our stacked model architecture can be compactly expressed using the `StackingRegressor` class in scikit-learn (*Pedregosa et al., 2011*) as of version 0.22.

# Acknowledgements

This work was partly supported by a 2018 'médecine numérique' (for digital medicine) thesis grant issued by Inserm (French national institute of health and medical research) and Inria (French national research institute for the digital sciences). It was also partly supported by the European Research Council Starting Grant SLAB ERC-StG-676943.

We thank Sheraz Khan for help with the Freesurfer segmentation and data management of the Cam-CAN dataset. We thank Mehdi Rahim for advice with the model stacking framework and data management of the Cam-CAN dataset. We thank Donald Krieger and Timothy Bardouille for help with the MEG co-registration. We thank Danilo Bzdok for feedback on the first version of the preprint.

# Additional information

### Competing interests

Gael Varoquaux: Reviewing editor, *eLife*. The other authors declare that no competing interests exist.

### Funding

| Funder | Grant reference number | Author |
| --- | --- | --- |
| H2020 European Research Council | SLAB ERC-StG-676943 | Alexandre Gramfort |
| Inria | Médecine Numérique 2018 | Denis A Engemann |
| Inserm | Médecine Numérique 2018 | Denis A Engemann |

The funders had no role in study design, data collection and interpretation, or the decision to submit the work for publication.

### Author contributions

Denis A Engemann, Conceptualization, Resources, Data curation, Software, Formal analysis, Supervision, Funding acquisition, Validation, Investigation, Visualization, Methodology, Writing - original draft, Project administration, Writing - review and editing; Oleh Kozynets, Resources, Data curation, Software, Investigation, Methodology, Writing - review and editing; David Sabbagh, Resources, Software, Writing - review and editing; Guillaume Lemaître, Resources, Software, Methodology, Writing - review and editing; Gael Varoquaux, Conceptualization, Formal analysis, Methodology, Writing - review and editing; Franziskus Liem, Methodology, Writing - review and editing; Alexandre Gramfort, Conceptualization, Software, Formal analysis, Supervision, Validation, Methodology, Project administration, Writing - review and editing

### Author ORCIDs

Denis A Engemann https://orcid.org/0000-0002-7223-1014
Gael Varoquaux http://orcid.org/0000-0003-1076-5122

## Ethics

Human subjects: This study is conducted in compliance with the Helsinki Declaration. No experiments on living beings were performed for this study. The data that we used was acquired by the Cam-CAN consortium and has been approved by the local ethics committee, Cambridgeshire 2 Research Ethics Committee (reference: 10/H0308/50).

## Decision letter and Author response

Decision letter https://doi.org/10.7554/eLife.54055.sa1
Author response https://doi.org/10.7554/eLife.54055.sa2

# Additional files

## Supplementary files

• Transparent reporting form

## Data availability

We used the publicly available Cam-CAN dataset (https://camcan-archive.mrc-cbu.cam.ac.uk/data-access/). All software and code necessary to obtain the derivative data is shared on GitHub: https://github.com/dengemann/meg-mri-surrogate-biomarkers-aging-2020 (copy archived at https://github.com/elifesciences-publications/meg-mri-surrogate-biomarkers-aging-2020).

The following previously published dataset was used:

| Author(s) | Year | Dataset title | Dataset URL | Database and Identifier |
| --- | --- | --- | --- | --- |
| Shafto MA, Tyler LK, Dixon M, Taylor JR, Rowe JB, Cusack R, Calder AJ, Marslen-Wilson WD, Duncan J, Dalgleish T, Henson RN, Brayne C, Matthews FE, Cam-CAN | 2014 | Cam-CAN | https://camcan-archive.mrc-cbu.cam.ac.uk/dataaccess/ | Cam-CAN Data Portal, Cam-CAN |

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
