## [Decision Letter]

**Acceptance summary:**

The authors describe a novel machine learning approach-opportunistic prediction stacking-with the aim of combining multiple neuroimaging modalities into a single predictive model (“biomarker”) in the face of missing data. Leveraging structural MRI, functional MRI, and MEG data from a large public biobank (Cam-Can), the authors show that each modality had additive incremental value in predicting age.

The reviewers and I were enthusiastic about the report. This approach has the potential to extract valuable information from multimodal databases and improve diagnostic power based on the surrogate biomarker idea. From a translational neuroscience perspective, the paper addresses (1) whether adding MEG improves age prediction and provides incremental predictive validity over sMRI or sMRI + fMRI alone; (2) which MEG features are most predictive of age. From a methodological perspective, the paper extends the original stacking method to address missing data, a common occurrence.

**Decision letter after peer review:**

Thank you for submitting your article "Combining electrophysiology with MRI enhances learning of surrogate-biomarkers" for consideration by *eLife*. Your article has been reviewed by two peer reviewers, and the evaluation has been overseen by Alex Shackman as the Reviewing Editor and Floris de Lange as the Senior Editor. The following individuals involved in review of your submission have agreed to reveal their identity: Kamen Tsvetanov (Reviewer #1); Nelson Trujillo-Barreto (Reviewer #2).

The reviewers have discussed the reviews with one another and the Reviewing Editor has drafted this decision to help you prepare a revised submission.

The authors describe a novel machine learning approach-opportunistic prediction stacking-with the aim of combining multiple neuroimaging modalities into a single predictive model (“biomarker”) in the face of missing data. Leveraging structural MRI, functional MRI, and MEG data from a large public biobank (Cam-Can), the authors show that each modality had additive incremental value in predicting age.

The reviewers and I were enthusiastic about the report:

• The paper is written and presented well.

• The research idea is interesting and timely and the proposed method has the potential to extract valuable information from multimodal databases and improve diagnostic power based on the surrogate biomarker idea.

• From a translational neuroscience perspective, the paper addresses (1) whether adding MEG improves age prediction and provides incremental predictive validity over sMRI or sMRI + fMRI alone; (2) which MEG features are most predictive of age.

• From a methodological perspective, the paper extends the original stacking method to address missing data, a common occurrence.

Challenges and Recommendations:

Nevertheless, our enthusiasm was somewhat tempered by several key limitations of the report. In this section, I briefly summarize the most important comments.

1) Challenge: – The motivation for including MEG seems to be exclusively based on EEG evidence. In the Introduction, all the evidence about the relationship between electrophysiology and ageing (and the complementarity to fMRI in that context) to motivate its inclusion as an additional modality is drawn from the EEG literature, which is odd, given that the paper uses MEG instead.

Recommendation: – The authors should motivate the use of MEG directly or at least link the evidence from EEG to the use of MEG in a more convincing way. This might be as simple as saying that, although a lot of evidence from EEG is available, there is no available EEG data in multimodal databases (which I think would be a fair statement in general), but given the relationship between the two techniques, it is expected that inclusion of MEG would be valuable as well, this link is not clear in the text.

2) Challenge: – The Materials and methods are not entirely clear. Some of the processing/feature extraction steps are not completely explained or appropriately justified. The taxonomy used for the extracted features in MEG is confusing.

Recommendation: – Clarify the Materials and methods.

3) Challenge: – Missing Statistics. – Statistical significance of some of the results are missing: Many of the results do not report any statistical significance threshold (or p-value), which makes claims about results being above chance, not rigorously justified. (e.g. "…All stacking models performed consistently better than chance…" What does it mean to perform better than chance here? Was a statistical significance test carried out? What was the null hypothesis tested? If so, report the p-value or equivalent used? In some cases, statistical significance is obvious, in others, it is difficult to assess via visual inspection.

Recommendation: – Report statistical significance of any comparisons made (either corrected or not) in the main report and in the supplement e.g. for MAE differences and MAE PE correlations.

4) Challenge: – Generality. – It is important to demonstrate the robustness and reliability of these features to generalize in unseen data.

Recommendation: – The authors could readily address this in the available data by splitting the sample in half (while maintaining the age distribution and data missingness) and test how similar are the loadings of each feature across data splits. The process could be repeated multiple times (1000s) to create a distribution, which can be compared to the distribution from a permuted data.

5) Challenge: – Non-random Missing Data. – It is important to show that the approach is not susceptible to confounds in the missing data (non-random missingness; e.g. more missing data coming from older individuals, which “helps” to learn an age-related effects).

Recommendation: – This could be easily addressed e.g. by comparing model performance between two scenarios of missingness in the fully available dataset. In one scenario the missing data come from subjects with uniform age distribution and in the other scenario a bias in age selection is introduced, i.e. larger portion of missing data comes from older individuals.

6) Challenge: – Need to better integrate or segregate the main report and the supplement. Extensive text is dedicated to supplementary figures (e.g. Figure 2 and Figure 4 figure supplements).

Recommendation: – If the supplementary material so important for reaching the conclusions of the paper, perhaps it would be more helpful to include them in the body of the paper. Otherwise I would suggest the authors to move the relevant explanations to the supplements.

7) Challenge: – The authors demonstrate that their approach can identify variability in the detected signals with behavioral relevance. The authors investigated this question by testing relations between brain age residuals and cognitive measures orthogonalized wrt chronological age. It is interesting why the authors have adopted this modular process, which has faced some criticism, as opposed to a single level model that can flexibly accommodate all variables in a single model (Lindquist, Geuter, Wager and Caffo, 2019).

Recommendation: – Authors should at least address this choice in the paper. (Lindquist, et al., 2019)

8) Challenge: – The authors do not seem to have considered the inclusion of variables of not interest which may lead to spurious correlations.

Recommendation: – Variables such as gender, handedness, head motion, total grey matter and total intracranial volume should be controlled for to ensure the specificity of the findings. One way to address both points would be to evaluate a model with multiple predictors including age, cognitive variable (e.g. Cattell) and covariates of not interest, where the dependent variable is brain age. Then the statistics on the cognitive variable of interest, which would indicate unique contribution to predict brain age over and above chronological age and covariates (i.e. individual variability), can be reported in the format of Figure 3.

9) Challenge: – The previous comment raised the concern about covariates of no interest and their consideration in the post hoc analysis.

Recommendation: – It would be really worthwhile having the authors' view whether the approach can/should include covariates of no interest in Layer I. This could be particularly relevant if each modality is associated with "unique" covariates of no interest (e.g. head motion in fMRI, or empty room recording SNR in MEG). Though, I assume that the combination of multiple datasets will reduce this bias, which brings me to my next point.

10) Recommendation: – It would be a really worthwhile the authors could consider reporting what part of the data in Level I contributed to the overall model performance, i.e. what is the topography in each modality that matters (e.g. atrophy in frontal regions, connectivity in DMN etc.)?

11) Challenge: – Some of the results or explanations for the results in the Discussion are not intuitive or perhaps adequately explained in the context of the existent literature and the design of the analysis.

Recommendation: – Revise the Discussion to address this concern.

12) Challenge: – Limitations – Limitations of the methodology (i.e. its assumptions) or the choice of the features (i.e. static vs dynamic) are not adequately put in context or discussed.

Recommendation: – Authors need to, at least briefly, discuss about how potential assumption violations might affect the results.

---

## [Author Response]

Challenges and Recommendations:Nevertheless, our enthusiasm was somewhat tempered by several key limitations of the report. In this section, I briefly summarize the most important comments.1) Challenge: The motivation for including MEG seems to be exclusively based on EEG evidence. In the Introduction, all the evidence about the relationship between electrophysiology and ageing (and the complementarity to fMRI in that context) to motivate its inclusion as an additional modality is drawn from the EEG literature, which is odd, given that the paper uses MEG instead.Recommendation: The authors should motivate the use of MEG directly or at least link the evidence from EEG to the use of MEG in a more convincing way. This might be as simple as saying that, although a lot of evidence from EEG is available, there is no available EEG data in multimodal databases (which I think would be a fair statement in general), but given the relationship between the two techniques, it is expected that inclusion of MEG would be valuable as well, this link is not clear in the text.

We thank the reviewers for this suggestion. We have now reworked the arguments connecting MEG and EEG to yield a more coherent and stringent motivation and emphasizing their differences more concretely. We have now expanded our review on multimodal MRI/electrophysiology datasets. Note that we have also adjusted the scope of the title to refocus the paper more strongly on MEG.

The paper is now entitled “Combining magnetoencephalography with magnetic resonance imaging enhances learning of surrogate-biomarkers”.

Please find below one new passage from the Introduction that best captures the spirit of this particular revision:

“At this point, there are very few multimodal databases providing access to electrophysiology alongside MRI and fMRI. […] MEG is, therefore, an interesting modality in its own right for developing neuro-cognitive biomarkers while its close link with EEG may potentially open the door to translatable electrophysiology markers suitable for massive deployment with clinical EEG.”

2) Challenge: The Materials and methods are not entirely clear. Some of the processing/feature extraction steps are not completely explained or appropriately justified. The taxonomy used for the extracted features in MEG is confusing.Recommendation: Clarify the Materials and methods.

We thank the reviewers for having shared their concerns. We have worked through the list of issues and revised the manuscript.

3) Challenge: Missing Statistics. Statistical significance of some of the results are missing: Many of the results do not report any statistical significance threshold (or p-value), which makes claims about results being above chance, not rigorously justified. (e.g. "…All stacking models performed consistently better than chance…" What does it mean to perform better than chance here? Was a statistical significance test carried out? What was the null hypothesis tested? If so, report the p-value or equivalent used? In some cases, statistical significance is obvious, in others, it is difficult to assess via visual inspection.Recommendation: Report statistical significance of any comparisons made (either corrected or not) in the main report and in the supplement e.g. for MAE differences and MAE PE correlations.

We understand the reviewers desire for a numerical estimate of statistical significance. While p-values can in principle be readily computed in the current setting for the question whether a model performed better than chance, e.g., by permuting the labels, the situation is less clear for model comparisons targeting differences in performance between two models. Rejecting a null-hypothesis that differences between models are due to chance would require several, ie., many independent datasets. Here, we computed uncertainty estimates of paired differences using the repeated 10-fold cross validation. For reasons of computational tractability, we estimated chance-level prediction using a dummy regressor that predicts the average of the training-set target using the same cross-validation procedure and identical random seeds to ensure split-wise comparability with non-trivial models.

To provide a more compact summary of the performance distributions that may help the reader to assess statistical inference, we extracted additional summary statistics of the distributions of mean absolute error scores and its paired differences with a reference model, where appropriate. These summaries included the mean, the standard deviation, the 2.5 and 97.5 percentiles and the number of splits in which the model was better than the reference. We have reworked the main text to make explicit the statistical approach and supplemented our report of the main findings inside the main text with these numerical uncertainty statistics. The following new section in the Materials and methods section explains our position and our methodological approach:

“Statistical Inference

Rejecting a null-hypothesis regarding differences between two cross-validated models is problematic in the absence of sufficiently large unseen data or independent datasets: cross-validated scores are not statistically independent. […] It should be clear, however, that hypothesis-testing, here, provides a quantitative orientation that needs to be contextualized by empirical estimates of effect sizes and their uncertainty to support inference.”

Finally, in the light of the cumulative comments made regarding the analysis of prediction errors across age, we have reworked that analysis to be visually more clearly represented and better integrated into the main text as Figure 2—figure supplement 2. In the process, we have attenuated our conclusions to be more nuanced and made the attempt to formalize inference by using an ANOVA model with age group, model family and their interactions as factors.

4) Challenge: Generality. It is important to demonstrate the robustness and reliability of these features to generalize in unseen data.Recommendation: The authors could readily address this in the available data by splitting the sample in half (while maintaining the age distribution and data missingness) and test how similar are the loadings of each feature across data splits. The process could be repeated multiple times (1000s) to create a distribution, which can be compared to the distribution from a permuted data.

We thank the reviewers for this suggestion. Our principal method across analyses for assessing robustness and reliability of features was the model comparisons method. This approach allowed us to track changes in performance as semantically and logically related blocks of intercorrelated variables are included or excluded. We used cross-validation to obtain an asymptotically unbiased estimate of the expected generalization error and its uncertainty distributions. Note that cross-validation already implements the resampling procedure suggested in this comment with the difference that 90% of the data were used for training in each round and duplication of procedures is not necessary as the cross validation distribution is sufficient to obtain useful inferences (see point above). However, we strongly agree that while expectations and their uncertainties are captured by the visualization and the newly added summary statistics, the relative stability of the model rankings may not be obvious based on our previous reports. We have therefore extended the reports to visualize and quantify the out-of-sample stability of the model ranking across the testing-set splits, which has given rise to novel supplementary figures.

For the specific inspection of the MEG model, we have now reported additional results based on two alternative variable importance metrics: 1) The out-of-sample permutations assessing the average permutation importance across cross-validation splits, which may yield an estimate of variables importance that is less prone to overfitting, and, 2) MDI importance potentially sensitive to the conditional dependencies between the variables but more prone to overfitting and false negatives/false positives. These additional analyses suggested that the importance ranking was highly consistent across methods with intercorrelations above .9. (Spearman rank correlation).

We have now reported the additional results in new supplementary figures, Figure 2—figure supplement 1 and Figure 4—figure supplements 1 and 2.

5) Challenge: Non-random Missing Data. It is important to show that the approach is not susceptible to confounds in the missing data (non-random missingness; e.g. more missing data coming from older individuals, which “helps” to learn an age-related effects).Recommendation: This could be easily addressed e.g. by comparing model performance between two scenarios of missingness in the fully available dataset. In one scenario the missing data come from subjects with uniform age distribution and in the other scenario a bias in age selection is introduced, i.e. larger portion of missing data comes from older individuals.

We thank the reviewers for raising this point. We feel that we may not have stated clearly enough that the sensitivity of our method with regard to non-random missingness is not a bug but a feature. Our method, by design, necessarily learns from any non-random missingness, which can be desired or undesired in different contexts. We have now extended the related Results section to make this point explicit and proposed as a diagnostic instrument to detect non-random missingness by training the random forest from the input data only including zeros or missingness indicators. In our case, the resulting model performance was well aligned with the distribution of chance-level scores, suggesting that missing values were not related to aging:

“It is important to emphasize that if missing values depend on age, the opportunistic model inevitably captures this information, hence, bases its predictions on the non-random missing data. […] In the current setting, the model trained on missing data indicators performed at chance level (*Pr*_<*Chance*_ = 30.00%, *M* = 0.65, *SD* = 1.68, *P*_2.5,97.5_ = [–2.96,3.60]), suggesting that the missing values were not informative of age.”

6) Challenge: Need to better integrate or segregate the main report and the supplement – Extensive text is dedicated to supplementary figures (e.g. Figure 2 and Figure 4 figure supplements).Recommendation: If the supplementary material so important for reaching the conclusions of the paper, perhaps it would be more helpful to include them in the body of the paper. Otherwise I would suggest the authors to move the relevant explanations to the supplements.

We have now revised the presentation of the results to yield a clearer division of labor between main text and supplement. Each supplementary analysis is now summarized by one sentence in the main text while providing the detailed contextual discussion in the respective supplementary figure captions:

For Figure 2 supplements:

“This additive component also became apparent when considering predictive simulations on how the model actually combined MEG, fMRI and MRI (Figure 2—figure supplement 2) using two dimensional partial dependence analysis (Karrer et al., 2019; Hastie et al., 2005, chapter 10.13.2). Moreover, exploration of the age-dependent improvements through stacking suggest that stacking predominantly reduced prediction errors uniformly (Figure 2—figure supplement 3) instead of systematically mitigating brain age bias (Le et al., 2018; Smith et al., 2019). ”

For Figure 4 supplements:

“Moreover, partial dependence analysis (Karrer et al., 2019; Hastie et al., 2005, chapter 10.13.2) suggested that the Layer-II random forest extracted non-linear functions (Figure 4—figure supplement 3). Finally, the best stacked models scored lower errors than the best linear models (Figure 4—figure supplement 4), suggesting that stacking achieved more than mere variable selection by extracting non-redundant information from the inputs.”

7) Challenge: The authors demonstrate that their approach can identify variability in the detected signals with behavioral relevance. The authors investigated this question by testing relations between brain age residuals and cognitive measures orthogonalized wrt chronological age. It is interesting why the authors have adopted this modular process, which has faced some criticism, as opposed to a single level model that can flexibly accommodate all variables in a single model (Lindquist, Geuter, Wager and Caffo, 2019).Recommendation: Authors should at least address this choice in the paper. (Lindquist et al., 2019).8) Challenge: The authors do not seem to have considered the inclusion of variables of not interest which may lead to spurious correlations.Recommendation: Variables such as gender, handedness, head motion, total grey matter and total intracranial volume should be controlled for to ensure the specificity of the findings. One way to address both points would be to evaluate a model with multiple predictors including age, cognitive variable (e.g. Cattell) and covariates of not interest, where the dependent variable is brain age. Then the statistics on the cognitive variable of interest, which would indicate unique contribution to predict brain age over and above chronological age and covariates (i.e. individual variability), can be reported in the format of Figure 3.

We thank the reviewers for sharing this reference and suggesting extended deconfounding. In fact, our method was based on the discussion in Smith et al., 2019

(https://doi.org/10.1016/j.neuroimage.2019.06.017) and pragmatically motivated : modular methods may be simpler to explain. To go beyond a modular model, we also performed a joint model with polynomial confounds such that $score = brain_age + poly(age, 3) + error$ and then extracted the brain age coef, this time quantifying effects conditional on the confounders. Moreover, we have included the additional confounders gender, handedness and motion parameters in a third model.

Note that motion correction was already performed during fMRI-preprocessing and that MEG source localization took into account individual head geometry as well as potentially confounding environmental noise through whitening with the noise covariance obtained from empty room recordings. Likewise, following the work by Liem et al., 2017, we included total grey matter and total intracranial volume as important features of interest among the MRI-features.

We found that the alternative models did not affect our conclusions and observed that deconfounding seemed to even improve the effect sizes of the models.

We have reworked and extended the Materials and methods description, included the citation regarding modularity and reported the alternative regression models in the subsection “Analysis of brain-behavior correlation”.

“The brain age Δ was defined as the difference between predicted and actual age of the person BrainAgeΔ=agepred−age, such that positive values quantify overestimation and negative value underestimation. […] Following the work by Liem et al., 2017, we included total grey matter and total intracranial volume as important features of interest among the MRI-features.”

See Figure 3—figure supplement 4 and Figure 3—figure supplement 5.

9) Challenge: The previous comment raised the concern about covariates of no interest and their consideration in the post hoc analysis.Recommendation: It would be really worthwhile having the authors' view whether the approach can/should include covariates of no interest in Layer I. This could be particularly relevant if each modality is associated with "unique" covariates of no interest (e.g. head motion in fMRI, or empty room recording SNR in MEG). Though, I assume that the combination of multiple datasets will reduce this bias, which brings me to my next point.

We thank the reviewers for raising this interesting point. We are somewhat worried that including covariates at the first level may unnecessarily inflate the number of estimated parameters while, at the same time, leading to limited results due to the lack of expressiveness in the ridge model. However, including covariates in the second layer should be more promising as the number of variables will remain small and the Random Forest can learn arbitrarily deep interaction effects between covariates and brainage models. The change in performance and variable importance can then be used to assess the impact of confounds. This question is methodologically interesting and motivates a dedicated study in a more specialized journal. We once more thank the reviewers for stimulating that interesting direction of thinking.

10) Recommendation: It would be a really worthwhile the authors could consider reporting what part of the data in Level I contributed to the overall model performance, i.e. what is the topography in each modality that matters (e.g. atrophy in frontal regions, connectivity in DMN etc.)?

We thank the reviewer for this remark. Interpreting high-dimensional linear models by their parameters is not necessarily an easy task. Collinearity and noise can induce strong weights on features that are not intrinsically important. Showing weights maps in such high dimensional settings requires dedicated tools (ReNa https://dx.doi.org/10.1109/TPAMI.2018.2815524, etc;) leading to modifications of the predictive models used in layer-1 which were optimized for prediction but not interpretability. Adopting interpretability methods would necessarily lead to a parallel method pipeline whose integration would exceed the scope of the paper. We have admitted this important limitation in the new dedicated limitations section at the end of the Discussion.

“For the present study, we see four principal limitations: availability of data, interpretability, nonexhaustive feature-engineering and potential lack of generalizability due to the focus on MEG. […] We hope, nevertheless, that the insights from our work will stimulate studies investigating the link between MEG, fMRI and MRI across the life-span using an inference-oriented framework.”

11) Challenge: Some of the results or explanations for the results in the Discussion are not intuitive or perhaps adequately explained in the context of the existent literature and the design of the analysis.Recommendation: Revise the Discussion to address this concern.

We thank the reviewers for pointing out the issues concerning the interpretation of the results. We have worked through the issues and updated the manuscript accordingly.

12) Challenge: Limitations – Limitations of the methodology (i.e. its assumptions) or the choice of the features (i.e. static vs dynamic) are not adequately put in context or discussed.Recommendation: Authors need to, at least briefly, discuss about how potential assumption violations might affect the results.

We thank the reviewers for this suggestion. We have now included an explicit limitations section at the end of the Discussion passage.